 

# Morphine disinhibits glutamatergic input to VTA dopamine neurons and promotes dopamine neuron excitation

**Ming Chen, Yanfang Zhao, Hualan Yang, Wenjie Luan, Jiaojiao Song, Dongyang Cui, Yi Dong, Bin Lai, Lan Ma, Ping Zheng\***

State Key Laboratory of Medical Neurobiology, Collaborative Innovation Center for Brain Science, School of Basic Medical Sciences and Institutes of Brain Science, Fudan Univeristy, Shanghai, China

**Abstract** One reported mechanism for morphine activation of dopamine (DA) neurons of the ventral tegmental area (VTA) is the disinhibition model of VTA-DA neurons. Morphine inhibits GABA inhibitory neurons, which shifts the balance between inhibitory and excitatory input to VTA-DA neurons in favor of excitation and then leads to VTA-DA neuron excitation. However, it is not known whether morphine has an additional strengthening effect on excitatory input. Our results suggest that glutamatergic input to VTA-DA neurons is inhibited by GABAergic interneurons via GABAB receptors and that morphine promotes presynaptic glutamate release by removing this inhibition. We also studied the contribution of the morphine-induced disinhibitory effect on the presynaptic glutamate release to the overall excitatory effect of morphine on VTA-DA neurons and related behavior. Our results suggest that the disinhibitory action of morphine on presynaptic glutamate release might be the main mechanism for morphine-induced increase in VTA-DA neuron firing and related behaviors.

\*For correspondence: pzheng@shmu.edu.cn

**Reviewing editor**: Wolfram Schultz, University of Cambridge, United Kingdom

## Introduction

Morphine is a potent analgesic with high addictive potential. Morphine-induced addictive behaviors are strongly dependent on the activation of dopamine (DA) neurons of the ventral tegmental area (VTA) (*Wise, 1989*; *Gardner, 2011*; *Luscher and Malenka, 2011*). The hedonic response produced by the activation of VTA-DA neurons by morphine is a primary factor behind drug dependence.

One previously reported mechanism for morphine activation of VTA-DA neurons is the VTA-DA neuron disinhibition model (*Johnson and North, 1992*; *Kalivas, 1993*; *White, 1996*). Morphine first inhibits neighboring GABA inhibitory neurons, which shifts the balance between inhibitory and excitatory input to VTA-DA neurons in favor of excitation and then leads to the promotion of VTA-DA neuron excitation. It was also reported that the release of glutamate from least some of the glutamate terminals that synapsed on VTA-DA neurons was inhibited by opioids (*Manzoni and Williams, 1999*; *Margolis et al., 2005*). This inhibitory effect of opioids on glutamate release is puzzling (*Chartoff and Connery, 2014*) because one would expect morphine to produce rapid activation of VTA-DA neurons through both the inhibition of GABAergic input and the excitation of glutamatergic input. In addition, postsynaptic inhibition (*Ford et al., 2006*) or excitation (*Margolis et al., 2014*) in response to µ opioid receptor activation has also been reported in some VTA-DA neurons. However, the amplitude of the inhibitory outward currents produced by opioids in postsynaptic VTA-DA neurons in the study of Ford et al. was small (2.1 ± 1.5 pA in VTA-DA neurons projecting to the nucleus accumbens and 14 ± 4 pA in those projecting to the basolateral amygdaloid nucleus) (*Ford et al., 2006*), while only a small population of VTA-DA neurons (19%) showed depolarization or an increase in firing rate in response

**eLife digest** Morphine is one of the most commonly used drugs for the treatment of severe pain. It is derived from opium, which is extracted from poppies, and binds to the same receptors in the brain as the body's own naturally produced painkillers. As well as providing pain relief, morphine can act directly on the brain's reward system to trigger a state of euphoria, and can therefore be highly addictive.

One of the key components of the brain's reward circuit that morphine affects is called the ventral tegmental area (VTA). The activity of the VTA is regulated by the combined efforts of two groups of cells: excitatory glutamatergic neurons that increase VTA activity and inhibitory interneuronsthat reduce the activity of the VTA.

Morphine inhibits the interneurons, thereby allowing the glutamatergic neurons to activate the VTA. But does morphine also strengthen this excitatory input directly? By examining the effects of morphine on individual VTA neurons, Chen et al. show that the drug does indeed enhance the activity of the glutamatergic neurons. However, it does so indirectly by inhibiting another group of interneurons that would otherwise silence the glutamatergic neurons. This effect of morphine is dependent on the drug acting on a specific receptor type on the interneurons.

Chen et al. show that injecting a drug that blocks these receptors straight into the VTA of rats prevents morphine from increasing the animals' activity levels. It also prevents the animals from developing a preference for being in locations where they have previously received morphine. This suggests that morphine could primarily exert its pleasurable effects by preventing the glutamatergic neurons from being inhibited, and thus allowing them to activate the VTA neurons.

to opioids in the study of *Margolis et al. (2014)*. This low postsynaptic responsiveness to opioids was consistent with physiological and anatomical evidence that only a few VTA-DA neurons had μ receptors (*Ford et al., 2006*).

Here, we propose that morphine might have a strengthening, rather than an inhibitory, effect on glutamatergic input to VTA-DA neurons because it was reported that blockade of excitatory glutamatergic signaling in the VTA suppressed morphine-induced VTA-DA neuron activation (*Jalabert et al., 2011*) and related behaviors (*Kalivas and Alesdatter, 1993*; *Harris and Aston-Jones, 2003*; *Harris et al., 2004*). Moreover, the strengthening influence of morphine on glutamatergic input to VTA-DA neurons is more relevant to its promoting effect on VTA-DA neuron excitation. In this article, we report our observations of the effect of morphine on glutamatergic input to VTA-DA neurons using the whole-cell patch-clamp method, and the results of our study of its mechanism and functional consequences using electrophysiological, biochemical, optogenetic, pharmacological, and behavioral approaches.

## Results

### Morphine-induced increase in spontaneous firing frequency of VTA-DA neurons is dependent on glutamatergic input

We observed the effect of morphine (10 μM) on the spontaneous firing frequency of VTA-DA neurons in rats. We select 10 μM of morphine because this concentration is commonly used in in vitro experiments as it elicits a significant effect useful for further analyses (*Akaishi et al., 2000*), and the effect of morphine at this concentration can be essentially abolished by the opioid receptor antagonist naloxone (*Valentino and Dingledine, 1982*). VTA-DA neurons were identified based on $I_h$ currents (*Figure 1A*) and tyrosine hydroxylase (TH) staining (*Figure 1B*). Details of the identification method are given in the 'Materials and methods' section. Using original recordings of spontaneous firing (left panel of *Figure 2A*) and the time course of spontaneous firing (middle panel of *Figure 2A*) in VTA-DA neurons for comparison, we could see that morphine (10 μM) increased the frequency of spontaneous firing in VTA-DA neurons. The average frequency of spontaneous firing increased from $1.0 \pm 0.3$ Hz before to $1.4 \pm 0.4$ Hz for 10–15 min after morphine application (n = 6 cells from five rats, paired *t* test, p < 0.05, compared to control before morphine, right panel of *Figure 2A*). In order to determine the role of glutamatergic input in the morphine-induced increase in the spontaneous firing

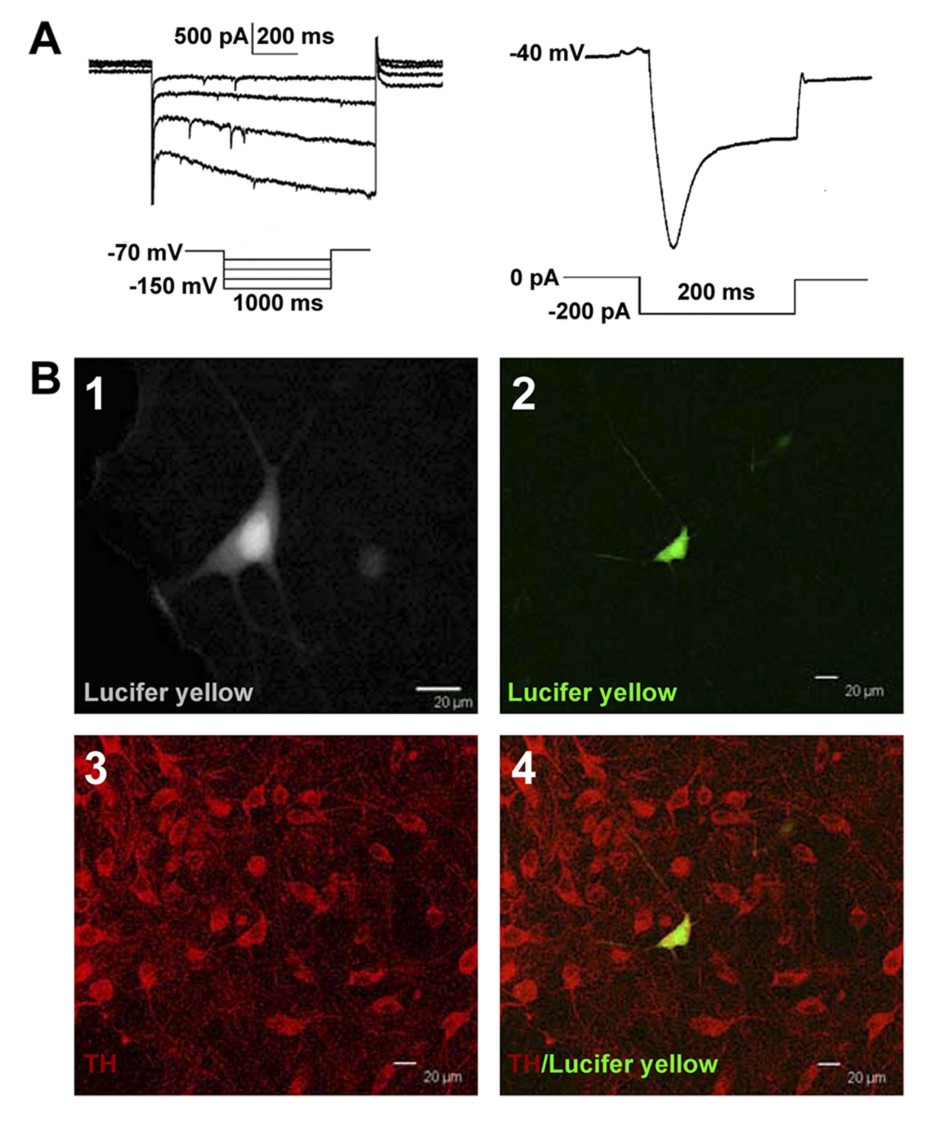

**Figure 1**. Identification of VTA-DA neurons in rats. (**A**) Electrophysiological properties of VTA-DA neurons. Left panel: representative traces showing a large hyperpolarization-activated current ($I_h$) in whole-cell voltage-clamp recording. Holding potential: −70 mV. Right panel: representative traces showing a large voltage 'sag' when hyperpolarized in whole-cell current-clamp recording. Holding current: 0 pA. (**B**) Immunohistochemical labeling of identified VTA-DA neurons. Panel 1: images of a Lucifer yellow-labeled neuron from the ventral tegmental area (VTA) after whole-cell patch-clamp recording under infrared differential interference contrast and fluorescent microscopy. Panel 2: the same neuron labeled with Lucifer yellow (green color) under confocal microscopy. Panel 3: VTA images showing tyrosine hydroxylase (TH)-positive neurons after immunostaining. Panel 4: Lucifer yellow-filled neuron co-labeled with TH. Scale bar: 20 μm.

frequency of VTA-DA neurons, we observed the influence of the N-methyl-D-aspartic acid (NMDA) receptor antagonist DL-2-amino-5-phosphonovaleric acid (APV) (50 μM) and the α-amino-3-hydroxy-5-methyl-4-isoxazolepropionic acid (AMPA) receptor antagonist 6, 7-Dinitroquinoxalie-2, 3-dione (DNQX) (10 μM) on the effect of morphine. In the presence of APV and DNQX, morphine no longer increased spontaneous firing frequency (*Figure 2B*). The average frequency of spontaneous firing was $0.7 \pm 0.1$ Hz before and $0.8 \pm 0.1$ Hz for 10–15 min after morphine application in the presence of APV and DNQX (n = 6 cells from five rats, paired *t* test, p > 0.05, compared to control with APV and DNQX before morphine, right panel of *Figure 2B*). These results suggest that the morphine-induced increase

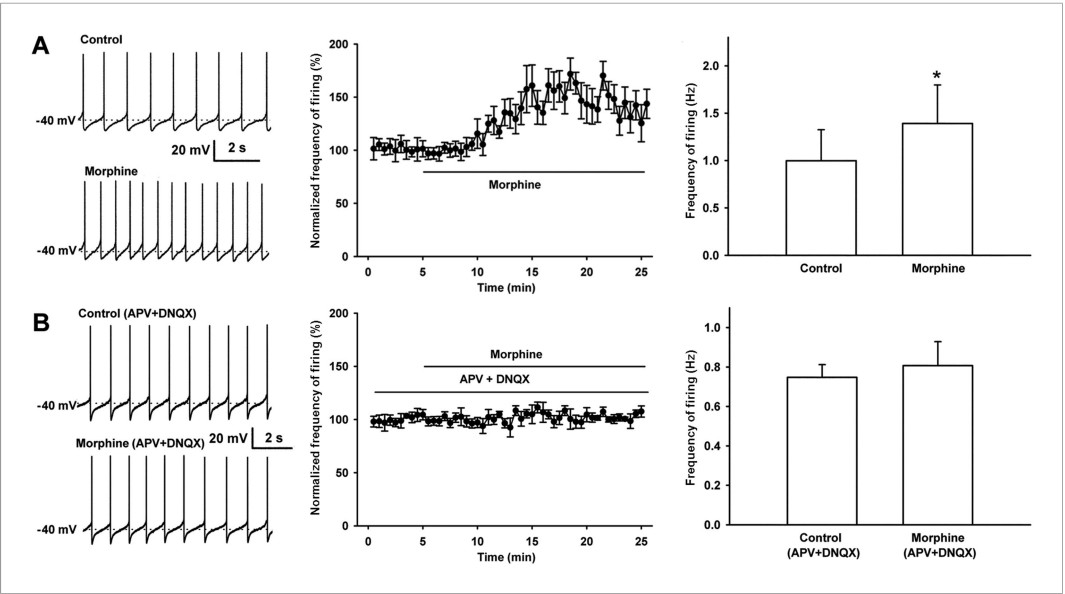

**Figure 2**. Effect of morphine on spontaneous firing of VTA-DA neurons in rats and the influence of the NMDA receptor antagonist APV and the AMPA receptor antagonist DNQX on the effect of morphine on spontaneous firing of VTA-DA neurons in rats. (**A**) Effect of morphine on spontaneous firing of VTA-DA neurons. Left panel: representative spontaneous firing traces before and after morphine (10 μM). Middle panel: time course of spontaneous firing before and after morphine (10 μM) ($n = 6$ cells from five rats). Right panel: average frequency of spontaneous firing before and after morphine ($n = 6$ cells from five rats, $p < 0.05$, compared to control before morphine). (**B**) Influence of the NMDA receptor antagonist APV and the AMPA receptor antagonist DNQX on the effect of morphine on spontaneous firing in VTA-DA neurons. Left panel: representative spontaneous firing traces before and after morphine (10 μM) in the presence of APV (50 μM) and DNQX (10 μM). Middle panel: time course of spontaneous firing before and after morphine in the presence of APV (50 μM) and DNQX (10 μM) ($n = 6$ cells from five rats). Right panel: average frequency of spontaneous firing before and after morphine in the presence of APV (50 μM) and DNQX (10 μM) ($n = 6$ cells from five rats, $p = 0.34$). Data are shown as the mean ±s.e.m. *$p < 0.05$.

in the spontaneous firing frequency of VTA-DA neurons requires AMPA and NMDA receptor-mediated glutamatergic input, consistent with a recent report using the NMDA antagonist APV and the AMPA receptor antagonist 6-Cyano-7-nitroquinoxaline-2, 3-dione (CNQX) in in vivo experiments (*Jalabert et al., 2011*).

## Morphine has an additional promoting effect on presynaptic glutamate release in VTA-DA neurons

In order to study the effect of morphine on glutamatergic input to VTA-DA neurons, we examined the effect of morphine on the frequency of spontaneous excitatory postsynaptic currents (sEPSCs) in VTA-DA neurons in rats. First, we observed the effect of morphine on the frequency of sEPSCs when the GABA$_A$ receptor antagonist picrotoxin (PTX) was added to a bath solution to remove spontaneous inhibitory postsynaptic currents (sIPSCs). Consistent with earlier reports (*Manzoni and Williams, 1999*; *Margolis et al., 2005*), in the presence of extracellularly applied PTX (100 μM), morphine (10 μM) decreased the frequency of sEPSCs (*Figure 3A*). The average frequency of sEPSCs decreased from $4.2 \pm 0.7$ Hz before to $3.5 \pm 0.7$ Hz for 10–15 min after morphine application ($n = 6$ cells from four rats, paired $t$ test, $p < 0.05$, compared to control before morphine, right panel of *Figure 3A*). However, bath application of GABA$_A$ receptor antagonists can lead to a wide blocking effect on GABA$_A$ receptors present on different types of neurons, including DA-neurons and GABA neurons, in VTA slices. To circumvent this wide influence and target the inhibition of GABA$_A$ receptors to VTA-DA neurons, we added the GABA$_A$ receptor antagonist PTX (100 μM) to the internal solution of microelectrodes as described by *Akaike et al. (1985)* when recording sEPSCs. To demonstrate the effectiveness of the blockade of GABA$_A$ receptors by intracellularly applied PTX, we first examined

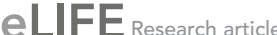

**Figure 3**. Effects of morphine on the frequency of spontaneous excitatory postsynaptic currents (sEPSCs) and paired pulse facilitation (PPF) of VTA-DA neurons in rats. (**A**) Effects of morphine on the frequency of sEPSCs in the presence of extracellularly applied picrotoxin (PTX) in VTA-DA neurons. Left panel: typical current traces of sEPSCs before and after morphine (10 µM) in the presence of extracellularly applied PTX. Middle panel: typical time course of the frequency of sEPSCs before and after morphine (10 µM) in the presence of extracellularly applied PTX. Right panel: average frequency of sEPSCs before and after morphine (10 µM) in the presence of extracellularly applied PTX (*n* = 6 cells from four rats, p < 0.05, compared to control before morphine). (**B**) Effects of morphine on sEPSCs in the presence of intracellularly applied PTX in VTA-DA neurons. Top of panel 1: inhibitory postsynaptic currents (IPSCs) in the normal intracellular recording solution and the presence of intracellularly applied PTX. Bottom of panel 1: typical current traces of sEPSCs before and after DNQX (10 µM) in the presence of intracellularly applied PTX. Panel 2: typical current traces of sEPSCs before and after morphine (10 µM) in the presence of intracellularly applied PTX.
*Figure 3. continued on next page*

*Figure 3. Continued*

Panel 3: typical time course of the frequency of sEPSCs before and after morphine (10 μM) in the presence of intracellularly applied PTX. Panel 4: average frequency of sEPSCs before and after morphine (10 μM) in the presence of intracellularly applied PTX (*n* = 8 cells from five rats, p < 0.05, compared to control before morphine). Panel 5: average amplitude of sEPSCs before and after morphine (10 μM) in the presence of intracellularly applied PTX (*n* = 8 cells from five rats, p = 0.24, compared to control before morphine). Panel 6: typical current traces of sEPSCs before and after DNQX (10 μM) in the presence of intracellularly applied PTX and morphine. (**C**) Effects of morphine on the PPF in VTA-DA neurons. Left panel: representative traces of the PPF before and after morphine (10 μM), and superimposition of the two traces normalized to the first excitatory postsynaptic current (EPSC) before and after morphine (10 μM) in the presence of intracellularly applied PTX. Middle panel: average amplitude of the first EPSC in control and morphine (10 μM) in the presence of intracellularly applied PTX (*n* = 6 cells from four rats, p < 0.05, compared to control before morphine). Right panel: average PPF before and after morphine (10 μM) in the presence of intracellularly applied PTX (*n* = 6 cells from four rats, p < 0.05, compared to control before morphine). (**D**) Effects of morphine on the frequency of sEPSCs when VTA-DA neurons were clamped the membrane potential at the reversal potential of Cl⁻ channels in VTA-DA neurons. Left panel: typical current traces of sEPSCs before and after morphine (10 μM) when DA neurons was clamped the membrane potential at the reversal potential of Cl⁻ channels. Middle panel: typical time course of the frequency of sEPSCs before and after morphine (10 μM) when DA neurons was clamped the membrane potential at the reversal potential of Cl⁻ channels. Right panel: average frequency of sEPSCs before and after morphine (10 μM) when DA neurons was clamped the membrane potential at the reversal potential of Cl⁻ channels (*n* = 6 cells from four rats, p < 0.05, compared to control before morphine). Data are shown as the mean ±s.e.m. *p < 0.05.

the effect of intracellularly applied PTX (100 μM) on inhibitory postsynaptic currents (IPSC) in VTA-DA neurons. The results showed that under this experimental paradigm, IPSCs disappeared (top of panel 1 in *Figure 3B*, n = 5 cells from two rats), demonstrating that GABA$_A$ receptors in the recorded VTA-DA neurons were blocked by intracellularly applied PTX. In addition, to further confirm that the spontaneous events we measured in the presence of intracellularly applied PTX were in fact sEPSCs, we observed the effect of the AMPA receptor antagonist DNQX on spontaneous events in the presence of intracellularly applied PTX. The results showed that the spontaneous events were completely blocked by DNQX (10 μM) (bottom of panel 1 in *Figure 3B*). We repeated the experiment in three cells from different slices and obtained similar results. On this basis, we observed the effect of morphine on the sEPSCs in the presence of intracellularly applied PTX. Raw current traces (panel 2 of *Figure 3B*) and the time course of sEPSCs (panel 3 of *Figure 3B*) before and after morphine application in the presence of intracellularly applied PTX showed that morphine (10 μM) increased the frequency of sEPSCs. The average frequency of sEPSCs increased from 4.8 ± 0.6 Hz before to 5.5 ± 0.6 Hz for 10–15 min after morphine application (n = 8 cells from five rats, paired *t* test, p < 0.05, compared to control before morphine, panel 4 of *Figure 3B*). However, morphine (10 μM) had no significant effect on the amplitude of sEPSCs. The average amplitude of sEPSCs was 16.6 ± 1.3 pA before and 15.3 ± 1.3 pA for 10–15 min after morphine application (n = 8 cells from five rats, paired *t* test, p > 0.05, compared to control before morphine, panel 5 of *Figure 3B*). To confirm that the increased spontaneous events we measured in the presence of intracellularly applied PTX after morphine were also in fact sEPSCs, we observed the effect of the AMPA receptor antagonist DNQX on spontaneous events after morphine application in the presence of intracellularly applied PTX. The results showed that the spontaneous events after morphine application were completely blocked by adding DNQX (10 μM) (panel 6 of *Figure 3B*). We repeated the experiment in three cells from different slices and obtained similar results. We also used the first excitatory postsynaptic current (EPSC) of paired pulse facilitation (PPF) as an index of EPSC (*Maejima et al., 2001*), and the PPF of paired EPSC as an indicator of presynaptic glutamate release (*Zucker and Regehr, 2002*) to confirm the effect of morphine on presynaptic glutamate release in VTA-DA neurons in the presence of intracellularly applied PTX. As shown in the left panel of *Figure 3C*, morphine (10 μM) increased the amplitude of the first EPSC, which was accompanied by a clear change in the presynaptic parameter PPF. The average amplitude of the first EPSCs was 124.1 ± 9.0 pA before and 161.9 ± 10.8 pA for 10–15 min after morphine application (n = 6 cells from four rats, paired *t* test, p < 0.05, compared to control before morphine, middle panel of *Figure 3C*). The average PPF was decreased from 1.5 ± 0.2 before to 1.2 ± 0.1 for 10–15 min after morphine application (n = 6 cells from four rats, paired *t* test,

p < 0.05, compared to control before morphine, right panel of *Figure 3C*). These results supported the suggestion that morphine increased presynaptic glutamate release in VTA-DA neurons in the presence of intracellularly applied PTX. More importantly, when we clamped the membrane potential of VTA-DA neurons at the reversal potential of Cl⁻ channels to remove sIPSCs, as an alternative to the application of a GABA$_A$ antagonist, either by the intracellular or bath approach, morphine still exerted a promoting effect on presynaptic glutamate release in VTA-DA neurons (*Figure 3D*). The average frequency of sEPSCs in this condition increased from 4.4 ± 0.3 Hz before to 5.5 ± 0.4 Hz for 10–15 min after morphine application (n = 6 cells from four rats, paired *t* test, p < 0.05, compared to control before morphine, right panel of *Figure 3D*).

## Morphine promotes presynaptic glutamate release in VTA-DA neurons via presynaptic disinhibition

The mechanism underlying the promoting effect of morphine on presynaptic glutamate release in VTA-DA neurons may involve different processes. One suggestion is that morphine may directly act at glutamatergic terminals to promote glutamate release. To test this hypothesis, we studied the effect of morphine on glutamate release from the VTA synaptosomes of rats, which are sealed particles containing vesicles, viable mitochondria, and all components necessary to store, release, and retain neurotransmitters (*Breukel et al., 1997*), using on-line fluorometry. However, we did not find that morphine (10 μM) had a direct effect at glutamatergic terminals to promote glutamate release. The average concentration of glutamate was 8.1 ± 0.1 nmol/mg before and 8.5 ± 0.7 nmol/mg after morphine application (n = 6 samples from eight rats, paired *t* test, p > 0.05, *Figure 4A*). We also studied the effect of morphine on the frequency of sEPSCs from mechanically dissociated single VTA-DA neurons in rats, which retained functional synaptic terminals (*Akaike and Moorhouse, 2003*; *Ye et al., 2004*; *Deng et al., 2009*). The left top image of panel 1 of *Figure 4B* shows mechanically

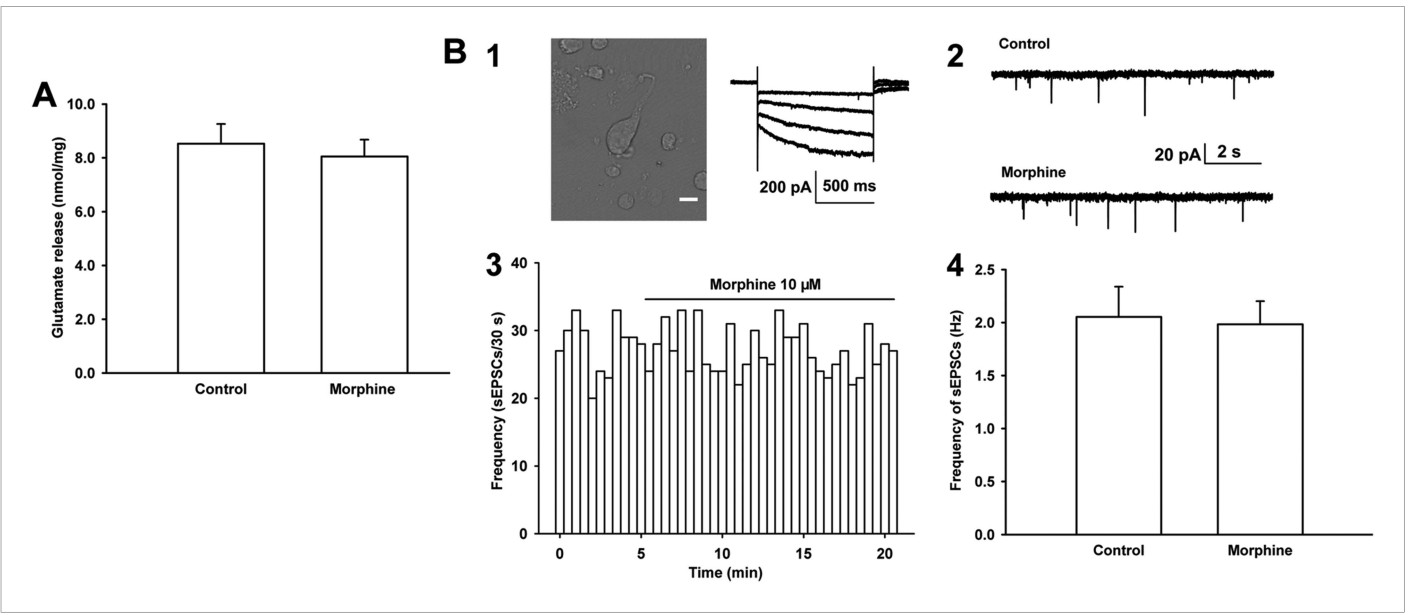

**Figure 4**. Effects of morphine on glutamate release from the ventral tegmental area (VTA) synaptosomes of rats and the frequency of sEPSCs in mechanically dissociated VTA-DA neurons from rats. (**A**) Effects of morphine on glutamate release from VTA synaptosomes. Average concentration of glutamate release before and after application of morphine (10 μM) from VTA synaptosomes (n = 6 samples from eight rats, p = 0.46). (**B**) Effects of morphine on the frequency of sEPSCs in mechanically dissociated VTA-DA neurons. Panel 1, left: images of an acutely dissociated single neuron from the VTA under phase contrast microscopy. Scale bar: 10 μm. Panel 1, right: representative current traces showing a large hyperpolarization-activated current (I$_h$) in whole-cell voltage-clamp recording. Holding potential: −70 mV. Panel 2: typical current traces of sEPSC before and after morphine (10 μM) in the presence of intracellularly applied PTX. Panel 3: typical time course of the frequency of sEPSCs before and after morphine (10 μM) in the presence of intracellularly applied PTX. Panel 4: Average frequency of sEPSCs before and after morphine (10 μM) in the presence of intracellularly applied PTX (n = 6 cells from six rats, p = 0.65). Data are shown as the mean ±s.e.m. *p < 0.05.

dissociated neurons from the VTA. These VTA-DA neurons were identified by only using $I_h$ currents (the graph on the right of panel 1 of *Figure 4B*) without TH staining because in dissociated DA neurons, it was difficult to fix the cell after recording for TH staining. The results showed that morphine (10 μM) had no significant effect on the frequency of sEPSCs (panels 2 and 3 of *Figure 4B*). The average frequency of sEPSCs was 2.1 ± 0.3 Hz before and 2.0 ± 0.2 Hz for 10–15 min after morphine application (10 μM) (n = 6 cells from six rats, paired *t* test, p > 0.05, compared to control before morphine, panel 4 of *Figure 4B*). These results suggest that morphine may promote presynaptic glutamate release in VTA-DA neurons in an indirect way.

To explore how morphine promoted presynaptic glutamate release in VTA-DA neurons, we hypothesized that glutamatergic input to VTA-DA neurons was inhibited by GABAergic interneurons and morphine disinhibited glutamatergic input by removing this inhibition, thus promoting glutamate release. To test this hypothesis, we first studied whether GABA could inhibit presynaptic glutamate release in VTA-DA neurons by examining the effect of exogenous applied of GABA on the frequency of sEPSCs of VTA-DA neurons in rats. From raw current traces (left panel of *Figure 5A*) and the time course of sEPSCs (middle panel of *Figure 5A*), we could see that GABA (10 μM) apparently decreased the frequency of sEPSCs. The average frequency of sEPSCs decreased from 6.7 ± 0.7 Hz before to 4.7 ± 0.6 Hz for 10–15 min after GABA application (n = 6 cells from four rats, paired *t* test, p < 0.05, compared to control before GABA, right panel of *Figure 5A*). Then, we observed whether the activation of intrinsic GABAergic neurons could inhibit presynaptic glutamate release of VTA-DA neurons in mice. To do this, AAV virus expressing a double floxed-stopped channelrhodopsin-2 (ChR2)-mCherry was stereotaxically injected into the VTA of mice expressing Cre recombinase in GABA neurons. 2 weeks after infection, expression of ChR2–mCherry was observed in the VTA (panel 1 of *Figure 5B*). We then performed whole-cell patch-clamp recording in GABA neurons of the VTA and observed the light-induced action potentials in GABA neurons in order to confirm that ChR2 was indeed expressed in the GABA neurons of the VTA in mice. The results showed that 470 nm light stimulation (20 Hz) elicited action potentials in GABA neurons (panel 2 of *Figure 5B*). On this basis, whole-cell patch-clamp recording was performed in VTA-DA neurons to observe the effect of 470 nm light stimulation on the frequency of sEPSCs in the presence of intracellularly applied PTX (100 μM). Following 5–10 min of baseline recording of sEPSCs, twenty 470 nm light pulses of 5 ms at 20 Hz were delivered every 4 s for 15 min. Raw current traces (panel 3 of *Figure 5B*) and the time course of sEPSCs (panel 4 of *Figure 5B*) before and after light stimulation showed that the light stimulation apparently decreased the frequency of sEPSCs. The average frequency of sEPSCs decreased from 4.1 ± 0.7 Hz before to 2.7 ± 0.3 Hz for 10–15 min after light stimulation (n = 6 cells from five mice, paired *t* test, p < 0.05, compared to control before light stimulation, panel 5 of *Figure 5B*).

We also studied which kinds of GABA receptors (GABA$_A$ or GABA$_B$ receptors) mediated the decreasing effect of GABA on presynaptic glutamate release in VTA-DA neurons by examining the influence of the GABA$_A$ or GABA$_B$ receptor antagonist on the effect of GABA on the frequency of sEPSCs in rats. The results showed that the GABA$_A$ receptor antagonist PTX (100 μM) had no significant influence on the effect of GABA (*Figure 6A*). The average frequency of sEPSCs still decreased from 4.0 ± 0.3 Hz before to 3.0 ± 0.2 Hz for 10–15 min after GABA application in the presence of PTX (n = 6 cells from four rats, paired *t* test, p < 0.05, compared to PTX before GABA, right panel of *Figure 6A*). The percentage of GABA-produced response in the presence of PTX (−25.8 ± 3.8%) was not statistically significant (n = 6, independent *t* test, p > 0.05) compared to that without PTX (−29.2 ± 6.0%). However, in the presence of the GABA$_B$ receptor antagonist CGP54626, the effect of GABA on the frequency of sEPSCs disappeared (*Figure 6B*). The average frequency of sEPSCs was 3.8 ± 0.4 Hz before and 3.8 ± 0.4 Hz for 10–15 min after GABA application in the presence of CGP54626 (2 μM) (n = 6 cells from four rats, paired *t* test, p > 0.05, compared to CGP54626 before GABA, right panel of *Figure 6B*). These results suggest that it is GABA$_B$ receptors, rather than GABA$_A$ receptors, that mediate the decreasing effect of GABA on presynaptic glutamate release in VTA-DA neurons.

To study the role of GABA$_B$ receptors in intrinsic GABA-induced inhibition of presynaptic glutamate release in VTA-DA neurons, we observed the effect of the GABA$_B$ receptor antagonist CGP54626 on the 470 nm light-induced inhibition of the frequency of sEPSCs of VTA-DA neurons in mice. From raw current traces (left panel of *Figure 6C*) and the time course of sEPSCs (middle panel of *Figure 6C*) before and after the 470 nm light stimulation in the presence of CGP54626 (2 μM), we could see that the inhibitory effect of the 470 nm light stimulation on the frequency of sEPSCs

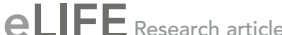

**Figure 5**. Effect of exogenous application of GABA and 470 nm light stimulation on the frequency of sEPSCs in the presence of intracellularly applied PTX in VTA-DA neurons. (**A**) Effect of exogenous application of GABA on the frequency of sEPSCs of VTA-DA neurons in rats. Left panel: typical current traces of sEPSCs before and after GABA (10 μM) in the presence of intracellularly applied PTX. Middle panel: typical time course of the frequency of sEPSCs before and after GABA (10 μM) in the presence of intracellularly applied PTX. Right panel: average frequency of sEPSCs before and after GABA (10 μM) in the presence of intracellularly applied PTX ($n = 6$ cells from four rats, $p < 0.05$, compared to control before GABA). (**B**) Effect of 470 nm light stimulation on the frequency of sEPSCs of VTA-DA neurons in mice. Panel 1: coronal image showing the expression of ChR2-mCherry (red) following injection of the viral construct bilaterally into the ventral tegmental area (VTA) of GADcre+ mice. Scale bar: 500 μm. Panel 2: 470 nm light (20 Hz)-induced firing of VTA GABA neurons in current-clamp mode. Panel 3: typical current traces of sEPSCs before and after blue light (470 nm) stimulation in the presence of intracellularly applied PTX. Panel 4: typical time course of the frequency of sEPSCs before and after blue light (470 nm) stimulation in the presence of intracellularly applied PTX. Panel 5: average frequency of sEPSCs before and after blue light (470 nm) stimulation in the presence of intracellularly applied PTX ($n = 6$ cells from five mice, $p < 0.05$, compared to control before light stimulation). Data are shown as the mean ±s.e.m. *$p < 0.05$.

disappeared in the presence of CGP54626. The average frequency of sEPSCs was $4.2 \pm 0.3$ Hz before and $4.1 \pm 0.2$ Hz for 10–15 min after 470 nm light stimulation in the presence of CGP54626 (n = 6 cells from five mice, paired $t$ test, $p > 0.05$, compared to control before 470 nm light stimulation in the presence of CGP54626, right panel of *Figure 6C*).

We examined the expression of GABA$_B$ receptors in the presynaptic terminals of the VTA in rats. Western blotting indicated that GABA$_B$ receptors were present in synaptosomes from the VTA (*Figure 7A*). Immunohistochemistry results for GABA$_B$ receptors, VGLUT2 (a glutamatergic terminal

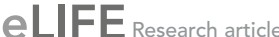

**Figure 6.** Influence of the GABA$_A$ receptor antagonist PTX and the GABA$_B$ receptor antagonist CGP54626 on the effect of exogenous application of GABA as well as the influence of the GABA$_B$ receptor antagonist CGP54626 on 470 nm light-induced inhibition of the frequency of sEPSCs in VTA-DA neurons. (**A**) Influence of the GABA$_A$ receptor antagonist PTX on the effect of exogenous application of GABA of VTA-DA neurons in rats. Left panel: typical current traces of sEPSC before and after GABA (10 µM) in the presence of PTX (100 µM). Middle panel: typical time course of the frequency of sEPSCs before and after GABA (10 µM) in the presence of PTX (100 µM). Right panel: average frequency of sEPSCs before and after GABA (10 µM) in the presence of PTX (100 µM) ($n = 6$ cells from four rats, $p < 0.05$, compared to PTX before GABA). (**B**) Influence of the GABA$_B$ receptor antagonist CGP54626 on the effect of exogenous application of GABA of VTA-DA neurons in rats. Left panel: typical current traces of sEPSC before and after GABA (10 µM) in the presence of CGP54626 (2 µM). Middle panel: typical time course of the frequency of sEPSCs before and after GABA (10 µM) in the presence of CGP54626 (2 µM). Right panel: average frequency of sEPSCs before and after GABA (10 µM) in the presence of CGP54626 (2 µM) ($n = 6$ cells from four rats, $p = 0.87$). (**C**) Influence of the GABA$_B$ receptor antagonist CGP54626 on 470 nm light-induced inhibition of the frequency of sEPSCs of VTA-DA neurons in mice. Left panel: typical current traces of sEPSCs before and after blue light (470 nm) stimulation in the presence of CGP54626 (2 µM). Middle panel: typical time course of the frequency of sEPSCs before and after blue light (470 nm) stimulation in the presence of CGP54626 (2 µM). Right panel: average frequency of sEPSCs before and after blue light (470 nm) stimulation in the presence of CGP54626 (2 µM) ($n = 6$ cells from five mice, $p = 0.21$). Data are shown as the mean ±s.e.m. *$p < 0.05$.

marker) and TH (a DA neuron marker) showed that GABA$_B$ receptors (green color, panel 1 of *Figure 7B*), VGLUT2 (red color, panel 2 of *Figure 7B*) and TH-positive neurons (blue color, panel 3 of *Figure 7B*) were present in VTA slices and the coexpression of GABA$_B$ receptors and VGLUT2 (yellow

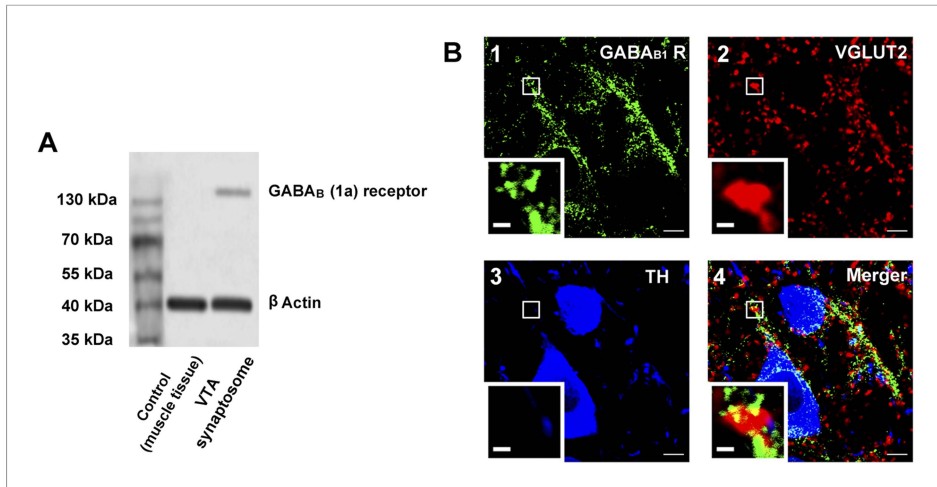

**Figure 7**. The presence of GABA_B receptors in the presynaptic glutamatergic terminals of the ventral tegmental area (VTA) in rats. (**A**) GABA_B receptor expression in the synaptosomes from the VTA. A representative Western blot shows GABA_B receptor expression in the synaptosomes from the VTA. (**B**) GABA_B receptor expression in the VTA shown using the triple-immunofluorescence method. Panel 1: GABA_B receptor immunolabeling (green-colored) in the VTA. Panel 2: VGLUT2-labeled axon terminals (red-colored) in the VTA. Panel 3: DA neuron labeled with TH (blue-colored) in the VTA. Panel 4: Merged image (yellow-colored) of GABA_B receptors and VGLUT2 in the VTA. The insets marked with small white rectangles in panels 1–4 are magnified views. Scale bars: 5 μm (in panels 1–4); 1 μm (in the insets in panels 1–4).

color, panel 4 of *Figure 7B*) indicated that GABA_B receptors were present in glutamatergic terminals of the VTA in rats. Moreover, this coexpression was close to TH-positive neurons (panel 4 of *Figure 7B*).

In addition, to determine whether there is basal GABAergic inhibitory control through GABA_B receptors on presynaptic glutamate release in VTA-DA neurons, we observed the effect of the GABA_B receptor antagonist CGP54626 on the frequency of sEPSCs in rats. The results showed that after the application of CGP54626, the frequency of sEPSCs significantly increased (*Figure 8A*). The average frequency of sEPSCs was $3.6 \pm 0.3$ Hz before and $4.5 \pm 0.3$ Hz for 10–15 min after CGP54626 application (2 μM) (n = 6 cells from four rats, paired $t$ test, $p < 0.05$, compared to control before CGP54626, right panel of *Figure 8A*). However, the GABA_A receptor antagonist PTX had no significant effect on the frequency of sEPSCs (*Figure 8B*). The average frequency of sEPSCs was $5.8 \pm 1.0$ Hz before and $5.7 \pm 0.9$ Hz for 10–15 min after PTX (100 μM) application (n = 6 cells from four rats, paired $t$ test, $p > 0.05$, compared to control before PTX, right panel of *Figure 8B*). This is consistent with the above result showing that GABA-mediated inhibition of presynaptic glutamate release in VTA-DA neurons is via GABA_B receptors rather than GABA_A receptors.

To study whether the morphine-induced increase in presynaptic glutamate release in VTA-DA neurons was via presynaptic disinhibition, we observed the effect of 'closing' local GABAergic interneurons on the morphine-induced increase in glutamate release in VTA-DA neurons using optogenetic methods in mice. AAV virus expressing a double floxed-stopped eNpHR3.0-EYFP was stereotaxically injected into the VTA of mice expressing Cre recombinase in GABA neurons. 2 weeks after infection, we performed whole-cell patch-clamp recording in GABA neurons of the VTA and observed light-induced inhibition of firing of action potentials in GABA neurons. The results showed that yellow light stimulation (590 nm) could reliably inhibit the current injection-induced firing of action potentials in GABA neurons (left panel of *Figure 9A*). On this basis, whole-cell patch-clamp recording was performed in VTA-DA neurons to observe the influence of the light-induced disinhibition of glutamatergic input on the effect of morphine. First, we observed whether this inhibition by light stimulation of GABA neurons affected the frequency of sEPSCs in VTA-DA neurons. As expected, light stimulation increased the frequency of sEPSCs in VTA-DA neurons (middle panel of *Figure 9A*). The average frequency of sEPSCs was $3.9 \pm 0.4$ Hz before and $4.5 \pm 0.4$ Hz for 10–15 min after light stimulation (n = 6 cells from five mice, paired $t$ test, $p < 0.05$, compared to control before light

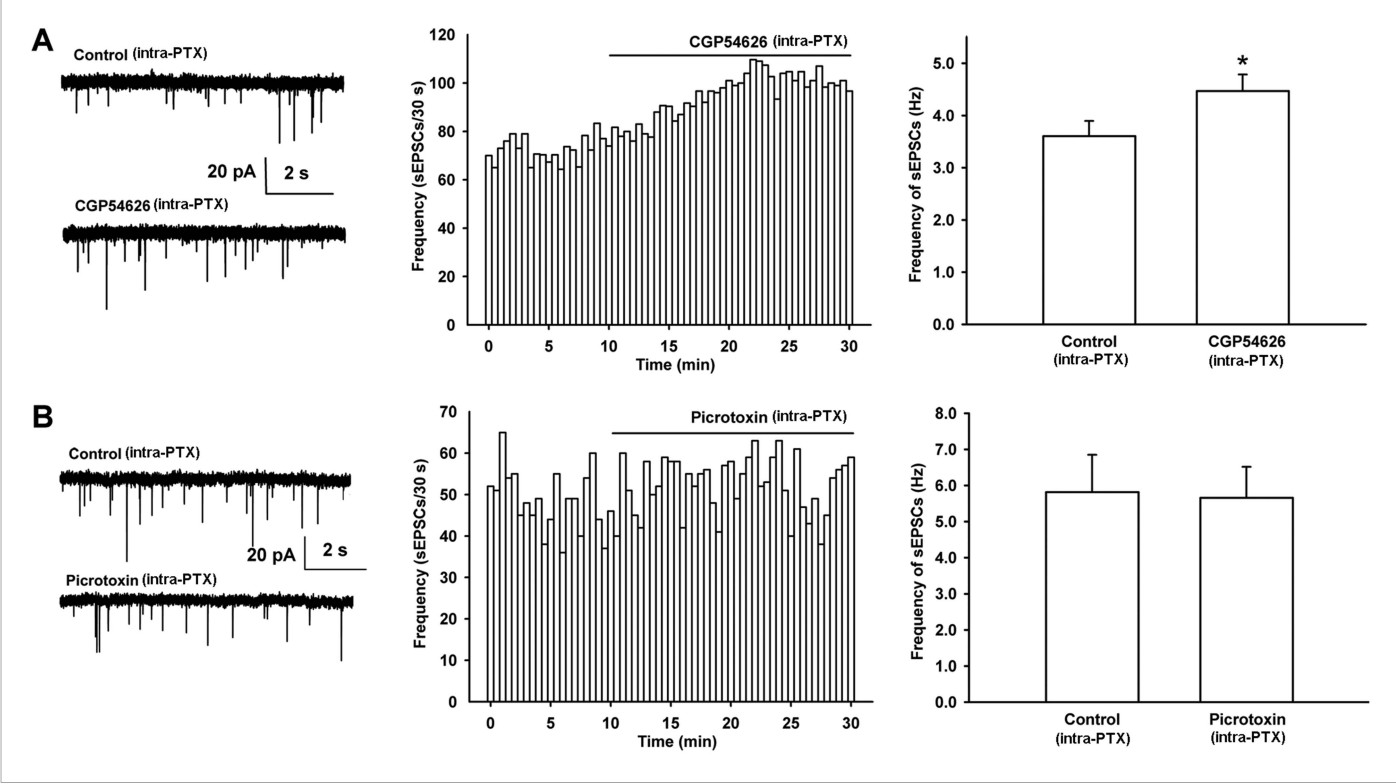

**Figure 8**. Effect of the GABA$_B$ receptor antagonist CGP54626 and the GABA$_A$ receptor antagonist PTX on the frequency of sEPSCs of VTA-DA neurons in rats. (**A**) Effect of the GABA$_B$ receptor antagonist CGP54626 on the frequency of sEPSCs in VTA-DA neurons. Left panel: typical current traces of sEPSCs before and after CGP54626 (2 µM). Middle panel: typical time course of the frequency of sEPSCs in DA neurons before and after CGP54626 (2 µM). Right panel: average frequency of sEPSCs before and after CGP54626 (2 µM) (n = 6 cells from four rats, p < 0.05, compared to control before CGP54626). (**B**) Effect of the GABA$_A$ receptor antagonist PTX on the frequency of sEPSCs in VTA-DA neurons. Left panel: typical current traces of sEPSCs before and after PTX (100 µM). Middle panel: typical time course of the frequency of sEPSCs before and after PTX (100 µM). Right panel: average frequency of sEPSCs before and after PTX (100 µM) (n = 6 cells from four rats, p = 0.29). Data are shown as the mean ±s.e.m. *p < 0.05.

stimulation, right panel of *Figure 9A*). Then, we observed the influence of the light-induced disinhibition of glutamatergic input on the effect of morphine. From raw current traces (left panel of *Figure 9B*) and the time course of sEPSCs (middle panel of *Figure 9B*), we could see that the effect of morphine on the frequency of sEPSCs disappeared in the presence of the light stimulation. The average frequency of sEPSCs was 4.5 ± 0.4 Hz before and 4.4 ± 0.5 Hz for 10–15 min after morphine application in the presence of the light stimulation (n = 6 cells from five mice, paired *t* test, p > 0.05, compared to control before morphine with light stimulation, right panel of *Figure 9B*). These results suggest that prior removal of the inhibition of GABAergic input on presynaptic glutamate release leads to disappearance of the effect of morphine on presynaptic glutamate release of VTA-DA neurons, indicating that morphine promotes presynaptic glutamate release of VTA-DA neurons via presynaptic disinhibition.

In addition, we used the GABA$_B$ receptor antagonist CGP54626 to remove inhibition of presynaptic glutamate release by GABAergic input and then observed its influence on the effect of morphine on glutamate release in rats. The results showed that after application of the GABA$_B$ receptor antagonist CGP54626 (2 µM), the frequency of sEPSCs significantly increased (*Figure 9C*). The average frequency of sEPSCs was 2.5 ± 0.4 Hz before and 3.0 ± 0.4 Hz for 5–10 min after CGP54626 application (n = 6 cells from four rats, paired *t* test, p < 0.05, compared to control before CGP54626, right panel of *Figure 9C*), suggesting that CGP54626 induced a disinhibitory effect on presynaptic glutamate release. On this basis, morphine (10 µM) was applied in the same cell, but did not further increase the frequency of sEPSCs (*Figure 9C*). The average frequency of sEPSCs was 3.0 ± 0.4 Hz before and 2.8 ± 0.4 Hz for 5–10 min after morphine application in the presence of

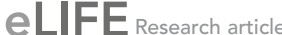

**Figure 9**. Effect of 590 nm light stimulation on the frequency of sEPSCs and influence of light-induced disinhibition of glutamatergic input and the GABA_B receptor antagonist CGP54626 on the effect of morphine on the frequency of sEPSCs in VTA-DA neurons. (**A**) Effect of 590 nm light stimulation on the frequency of sEPSCs of VTA-DA neurons in mice. Left panel: 590 nm light inhibits current injection-induced firing of action potentials in VTA GABA neurons. Middle panel: typical time course of the frequency of sEPSCs before and after yellow light (590 nm) stimulation. Right panel: average frequency of sEPSCs before and after yellow light (590 nm) stimulation ($n = 6$ cells from five mice, $p < 0.05$, compared to control before light stimulation).
(**B**) Influence of light-induced disinhibition of glutamatergic input on the effect of morphine on the frequency of sEPSCs of VTA-DA neurons in mice. Left panel: typical current traces of sEPSC before and after morphine (10 µM) in the presence of yellow light (590 nm) stimulation. Middle panel: typical time course of the frequency of sEPSCs before and after morphine (10 µM) in the presence of yellow light (590 nm) stimulation. Right panel: average frequency of sEPSCs before and after morphine in the presence of yellow light (590 nm) stimulation ($n = 6$ cells from five mice, $p = 0.78$). (**C**) Influence of the GABA_B receptor antagonist CGP54626 on the effect of morphine on the frequency of sEPSCs of VTA-DA neurons in rats. Left panel: typical current traces of sEPSCs before and after morphine (10 µM) in the presence of CGP54626 (2 µM). Middle panel: typical time course of the frequency of sEPSCs before and after morphine (10 µM) in the presence of CGP54626 (2 µM). Right panel: plots of the average frequency of sEPSCs in control, in CGP54626 (2 µM), and in morphine (10 µM) ($n = 6$ cells from four rats; $p < 0.05$, CGP54626 compared to control, $p < 0.05$, morphine compared to control). Data are shown as the mean ±s.e.m. *$p < 0.05$, #$p < 0.05$.

CGP54626 (n = 6 cells from four rats, paired *t* test, p > 0.05, compared to CGP54626 before morphine, right panel of *Figure 9C*).

## Contribution of the morphine-induced disinhibitory effect on presynaptic glutamate release in VTA-DA neurons to the overall excitatory effect of morphine on VTA-DA neurons

We studied the influence of the selective presynaptic GABA_B receptor antagonist CGP36216 on the effect of morphine on the frequency of spontaneous firing of VTA-DA neurons in rats. First, we observed the effect of CGP36216 on the frequency of spontaneous firing in VTA-DA neurons. The results showed that CGP36216 (100 µM) could increase the frequency of spontaneous firing in VTA-DA neurons (*Figure 10A*). The average frequency of spontaneous firing increased from 1.8 ± 0.3 Hz before to 2.1 ± 0.3 Hz after CGP36216 application (n = 6 cells from four rats, paired *t* test, p < 0.05, compared to control before CGP36216, right panel of *Figure 10A*). Then, we observed the influence of CGP36216 on the effect of morphine on the frequency of spontaneous firing in VTA-DA neurons. As shown by raw spontaneous firing traces (left panel of *Figure 10B*) and the time course of spontaneous firing (middle panel of *Figure 10B*) in VTA-DA neurons, the effect of morphine (10 µM) disappeared in the presence of CGP36216 (100 µM). The average frequency of spontaneous firing in

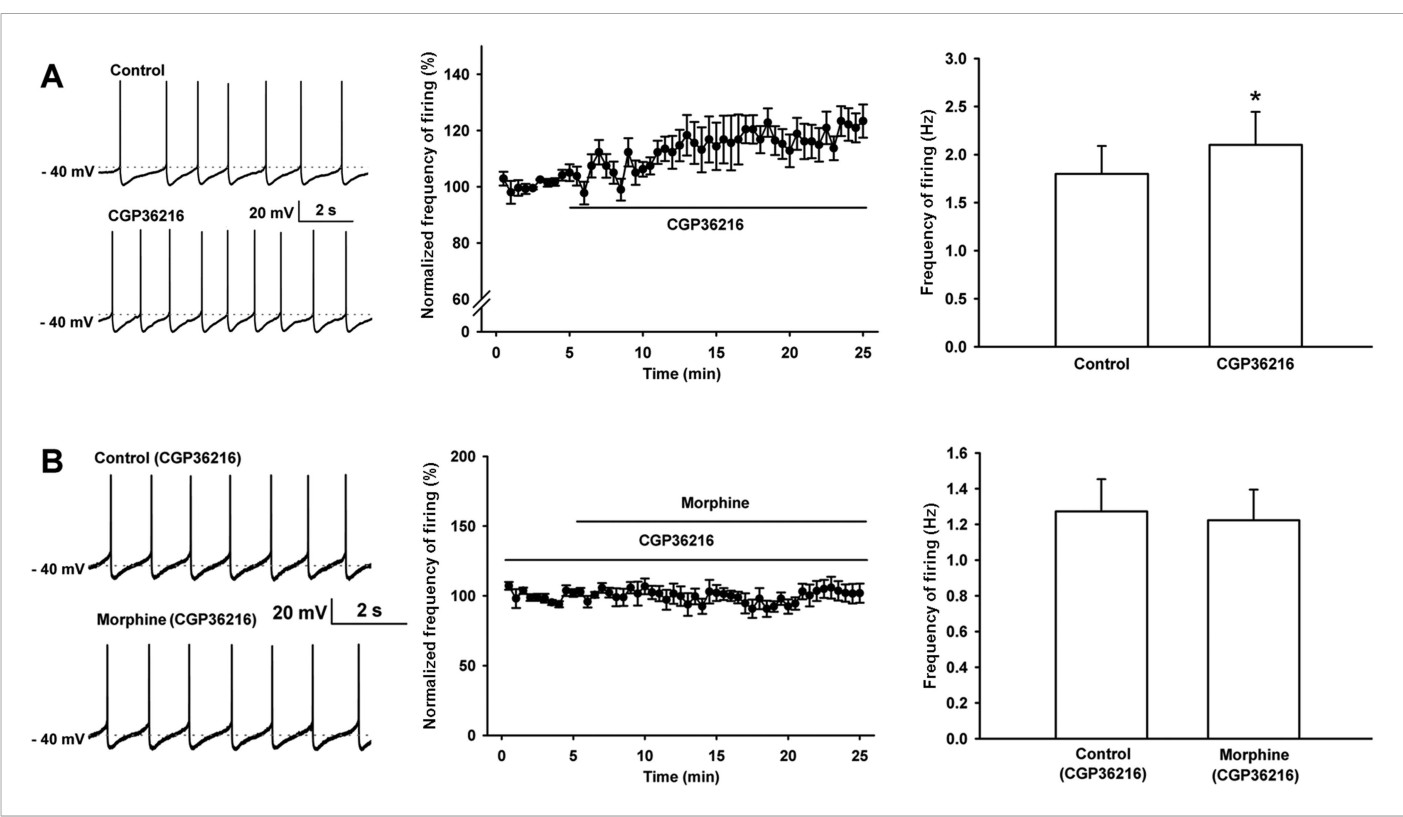

**Figure 10**. Effect of the selective presynaptic GABA_B receptor antagonist CGP36216 on spontaneous firing and the influence of the selective presynaptic GABA_B receptor antagonist CGP36216 on the effect of morphine on spontaneous firing of VTA-DA neurons in rats. (**A**) Effect of the selective presynaptic GABA_B receptor antagonist CGP36216 on spontaneous firing in VTA-DA neurons. Left panel: representative spontaneous firing traces before and after CGP36216 (100 µM). Middle panel: time course of spontaneous firing before and after CGP36216 (100 µM) (n = 6 cells from four rats). Right panel: average frequency of spontaneous firing before and after CGP36216 (100 µM) (n = 6 cells from four rats, p < 0.05, compared to control before CGP36216). (**B**) Influence of the selective presynaptic GABA_B receptor antagonist CGP36216 on the effect of morphine on spontaneous firing in VTA-DA neurons. Left panel: representative spontaneous firing traces before and after morphine (10 µM) in the presence of CGP36216 (100 µM). Middle panel: time course of spontaneous firing before and after morphine (10 µM) in the presence of CGP36216 (100 µM) (n = 6 cells from four rats). Right panel: average frequency of spontaneous firing before and after morphine (10 µM) in the presence of CGP36216 (100 µM) (n = 6 cells from four rats, p = 0.35, compared to CGP36216 before morphine). Data are shown as the mean ±s.e.m. *p < 0.05.

VTA-DA neurons was 1.3 ± 0.2 Hz before and 1.2 ± 0.2 Hz for 10–15 min after morphine application in the presence of CGP36216 (n = 6 cells from four rats, paired *t* test, p > 0.05, compared to CGP36216 before morphine, right panel of *Figure 10B*).

Since enhanced DA function in the VTA has most often been assessed as increased locomotor activity at the behavioral level (*Vezina, 2004*), we also studied the contribution of the morphine-induced disinhibition of glutamatergic input in the VTA to behavioral changes produced by morphine. In behavioral experiments, to avoid morphine acting on other brain areas sensitive to morphine, we injected morphine locally into the VTA in rats. Injection sites were verified under light microscope (left panel of *Figure 11A*); the injection sites used for data analysis are shown in the middle and right panels of *Figure 11A*. We first observed the effect of CGP36216 on locomotor activity. The results showed that CGP36216 (20 µg/rat) increased locomotor activity. The distance traveled by rats was 6.3 ± 1.4 m in the control group compared with 12.2 ± 2.0 m in the CGP36216 group (n = 6 rats, independent *t* test, p < 0.05). Then we observed the influence of CGP36216 on the effect of morphine on locomotor activity in rats. As shown in *Figure 11B*, locomotor activity was significantly increased following intra-VTA injection of morphine (1 µg/rat). The distance traveled by rats was 6.3 ± 1.4 m in the control group compared with 37.6 ± 9.6 m in the morphine alone group (n = 6 rats, independent *t* test, p < 0.05, right panel of *Figure 11B*). However, in animals with intra-VTA injection of CGP36216 (20 µg/rat), the morphine-induced increase in locomotor activity disappeared (*Figure 11B*). The distance traveled by rats was 37.6 ± 9.6 m in the morphine alone group compared with 7.5 ± 1.4 m in the morphine plus intra-VTA CGP36216 group (n = 6 rats, independent *t* test, p < 0.05, right panel of *Figure 11B*). In addition, to evaluate the role of VTA GABA_B receptors in the overall effect of intraperitoneal (i.p.) morphine, we observed the effect of intra-VTA injected CGP36216 on i.p. morphine-induced increase in locomotor activity. As shown in *Figure 11C*, locomotor activity was significantly increased by i.p. injection of morphine (10 mg/kg) plus intra-VTA injection of saline. The distance traveled by rats was 5.4 ± 0.8 m in the control group compared with 28.3 ± 5.7 m in the i.p. morphine (10 mg/kg) plus intra-VTA saline group (n = 6 rats, independent *t* test, p < 0.05, right panel of *Figure 11C*). However, in animals with intra-VTA injection of CGP36216 (20 µg/rat), the i.p. morphine-induced increase in locomotor activity disappeared (*Figure 11C*). The distance traveled by rats was 28.3 ± 5.7 m in the i.p. morphine (10 mg/kg) plus intra-VTA saline group compared with 6.6 ± 2.3 m in the i.p. morphine plus intra-VTA CGP36216 group (n = 6 rats, independent *t* test, p < 0.05, right panel of *Figure 11C*).

We also observed the effect of the presynaptic GABA_B receptor antagonist CGP36216 on morphine-induced conditioned place preference (CPP) in rats. Morphine (1 µg/rat) and CGP36216 (20 µg/rat) were locally injected into the VTA. Injection sites for data analysis are shown in *Figure 12A*. A schematic of the experimental design for CPP and drug application is shown in *Figure 12B*. As shown in *Figure 12C*, two-way ANOVAs conducted on the CPP score using treatment with different drugs as the between-subjects factors and test condition (preconditioning and postconditioning) as the within-subjects factor, revealed that there was a significant interaction of treatment ($F_{(3,40)}$ = 4.45; p = 0.008) and test condition ($F_{(1,40)}$ = 19.79; p < 0.001). Post-hoc analysis showed that after CPP training, the morphine group (n = 6 rats) exhibited greater CPP compared with the control group (n = 6 rats, p < 0.05) but in groups receiving CGP36216 (n = 6 rats), morphine-induced CPP was absent (p < 0.05). These results suggest that intra-VTA injection of a presynaptic GABA_B receptor antagonist abolishes CPP induced by intra-VTA injection of morphine.

## Discussion

Previous studies reported that µ receptor agonists decreased EPSCs (*Bonci and Malenka, 1999*; *Margolis et al., 2005*). In agreement, as shown by the results above, if a GABA_A receptor antagonist was applied using a similar bath approach, morphine inhibited the frequency of sEPSCs. However, if the GABA_A receptor antagonist was applied using an intracellular approach, morphine had a promoting rather than an inhibitory effect.

Why did morphine produce opposite effects on the frequency of sEPSCs under different GABA_A receptor antagonist application conditions? We believe that under the condition of bath application of a GABA_A receptor antagonist, the antagonist can produce a wide inhibitory effect on GABA_A receptors present on various neuron subtypes, such as DA neurons and GABA neurons, in VTA slices. Maybe this wide effect shifts the effect of morphine on presynaptic glutamate release in VTA-DA neurons toward an inhibitory effect. However, the intracellular application of a GABA_A receptor

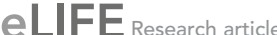

**Figure 11**. Influence of intra-ventral tegmental area (VTA) injection of the presynaptic GABA$_B$ receptor antagonist CGP36216 on morphine-induced increase in locomotor activity in rats. (**A**) Injection sites were verified under light microscope. Left panel: Representative Nissl-stained photomicrograph of cannula tracts terminating in the VTA. Right panel: location of the injection cannula tips in the VTA of rats used in data analyses. Numbers indicate coordinates relative to bregma. (**B**) Influence of intra-VTA injection of the presynaptic GABA$_B$ receptor antagonist CGP36216 on intra-VTA injection of morphine inducing an increase in locomotor activity. Left panel: time course of locomotor activity before and after intra-VTA injection of saline or morphine (1 µg/rat), or morphine (1 µg/rat) with CGP36216 (20 µg/rat) ($n = 6$ rats). Right panel: average distance traveled by rats during 120 min after treatment with an intra-VTA injection of saline or morphine (1 µg/rat), or morphine (1 µg/rat) with CGP36216 (20 µg/rat) ($n = 6$ rats; *$p < 0.05$, compared with intra-VTA injection of saline, #$p < 0.05$, compared with intra-VTA injection of morphine. (**C**) Influence of intra-VTA injection of the presynaptic GABA$_B$ receptor antagonist CGP36216 on intraperitoneal (i.p.) morphine-induced increase in locomotor activity. Left panel: time course of locomotor activity before and after intra-VTA injection of saline co-administered with either i.p. saline (1 ml/kg) or morphine (10 mg/kg), and i.p. morphine (10 mg/kg) with intra-VTA injection of CGP36216 (20 µg/rat) ($n = 6$ rats). Right panel: average distance traveled by rats during 120 min after treatment with intra-VTA injection of saline co-administered with either i.p. saline (1 ml/kg) or morphine (10 mg/kg), and i.p. morphine (10 mg/kg) with intra-VTA injection of CGP36216 (20 µg/rat) ($n = 6$; *$p < 0.05$, compared with intra-VTA injection of saline co-administered with i.p. saline, #$p < 0.05$, compared with intra-VTA injection of saline co-administered with i.p. morphine). Data are shown as the mean ±s.e.m.

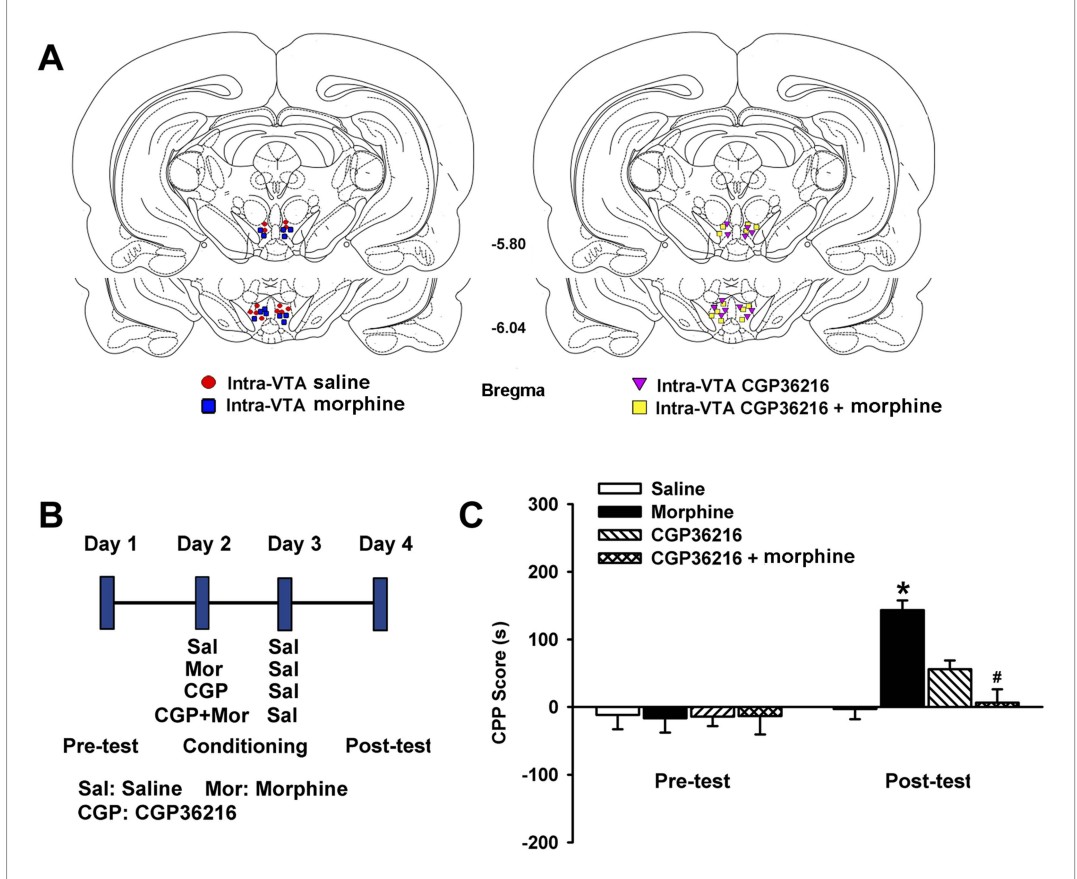

**Figure 12**. Influence of intra-ventral tegmental area (VTA) injection of the presynaptic GABA$_B$ receptor antagonist CGP36216 on intra-VTA injected morphine-induced conditioned place preference (CPP) in rats. (**A**) Schematic representations of injection cannula tips in the VTA of rats used in data analyses. Numbers indicate coordinates relative to bregma. (**B**) A schematic of the experimental design for CPP and administration of drugs. (**C**) Influence of the presynaptic GABA$_B$ receptor antagonist CGP36216 on morphine-induced CPP in rats. Averaged CPP score of preconditioning and postconditioning in different groups ($n = 6$ rats in the every group; *$p < 0.05$, intra-VTA injection of morphine group compared with intra-VTA injection of saline group, #$p < 0.05$, intra-VTA injection of morphine plus CGP36216 group compared with intra-VTA injection of morphine group). Data are shown as the mean ±s.e.m.

antagonist only blocks GABA$_A$ receptors in the single VTA-DA neuron being recorded. In this case, a clear promoting effect on presynaptic glutamate release in VTA-DA neurons was observed. In addition, to confirm this promoting effect under intracellularly applied PTX, we used the PPF of EPSC as another indicator of presynaptic glutamate release in order to observe the effect of morphine; the result substantiated our conclusion that morphine promoted presynaptic glutamate release in VTA-DA neurons. More importantly, when we clamped the membrane potential of VTA-DA neurons in the reversal potential of Cl⁻ channels to remove sIPSCs, instead of using the intracellular or bath application of a GABA$_A$ antagonist, morphine still promoted presynaptic glutamate release in VTA-DA neurons. These results strongly suggest that morphine can promote presynaptic glutamate release in VTA-DA neurons.

Postsynaptic inhibition in response to μ opioid receptor activation has been reported in some VTA-DA neurons (*Ford et al., 2006*). Surprisingly, we did not observe this effect. The amplitude of the outward currents produced by opioids in the study by Ford et al. was small (2.1 ± 1.5 pA in VTA-DA neurons projecting to the nucleus accumbens and 14 ± 4 pA in those projecting to the basolateral amygdaloid nucleus) and thus might not be reflected by the change in action potential firing in our study. In addition, a recent report demonstrated postsynaptic excitation of some VTA-DA neurons by

opioids (*Margolis et al., 2014*), an event that we also did not observe. Only a small population of VTA-DA neurons (19%) showed depolarization or an increase in firing rate in response to opioids in this paper, which is consistent with physiological and anatomical evidence that μ receptors were present on a small population of VTA-DA neurons (*Ford et al., 2006*). Consequently, we believe that we did not observe this effect perhaps because the percentage of the cells responding by excitation was low. Similar to our study, Jalabert et al. did not observe an excitatory effect of firing of VTA-DA neurons by opioids in the presence of glutamate receptor antagonists (*Jalabert et al., 2011*).

The mechanism underlying the promoting effect of morphine on presynaptic glutamate release in VTA-DA neurons may involve different processes. One suggestion is that morphine may directly act at glutamatergic terminals to promote glutamate release. To test this hypothesis, we observed the effect of morphine on glutamate release in isolated nerve terminals—synaptosomes (*Breukel et al., 1997*) from the VTA. We found that morphine had no significant effect on glutamate release. In addition, using mechanically dissociated single VTA-DA neurons, we still did not observe an effect of morphine on the frequency of sEPSCs. These results suggest that morphine does not exert a direct promoting effect on presynaptic glutamate release in VTA-DA neurons.

Another possible mechanism may be through a local neuronal circuit that promotes glutamate release. It has been proposed that μ receptors located in GABAergic interneurons are the primary site of action of opiates in the VTA (*Hyman et al., 2006*). This is consistent with morphological and functional data showing that μ receptors are primarily located on non-dopaminergic neurons in the VTA (*Garzon and Pickel, 2001*) and that activation of μ receptors can inhibit GABAergic interneurons by hyperpolarizing the cells (*Johnson and North, 1992*). In addition, Chefer et al. studied the inter-relationship of μ receptors, GABA, glutamate, and DA in the VTA of freely moving animals (*Chefer et al., 2009*). They found that there was a positive correlation between basal dopamine levels and the glutamate/GABA ratio in the VTA of wild-type animals. Moreover, in μ receptor knockout mice, the GABAergic inhibitory tone in the VTA was significantly increased, but the glutamatergic excitatory tone was decreased. These results suggest that activation of μ receptors may increase presynaptic glutamate release via inhibition of GABAergic interneurons in VTA-DA neurons. This is supported by our results indicating that the μ receptor antagonist CTOP can block the promoting effect of morphine on the frequency of sEPSCs (the average frequency of sEPSCs was $2.95 \pm 0.35$ Hz before and $2.97 \pm 0.40$ after morphine application (10 μM) in the presence of CTOP (1 μM); n = 6 cells from four rats, $p > 0.05$). Therefore, we hypothesize that morphine may act at μ receptors located in GABA neurons, inhibit the function of GABA neurons, and thus disinhibit the presynaptic glutamate release of VTA-DA neurons, leading to increased glutamate release.

To test whether morphine has a disinhibitory effect on the presynaptic glutamate release of VTA-DA neurons, the presence of an inhibitory circuit from neighboring GABA neurons to the presynaptic glutamatergic terminals of VTA-DA neurons must first be demonstrated. Therefore, we designed a series of experiments to determine if this inhibitory circuit existed. Our results showed that (1) GABA could inhibit the presynaptic glutamate release of VTA-DA neurons and this inhibition was mediated through GABA$_B$ receptors, rather than GABA$_A$ receptors; (2) GABA$_B$ receptors were present in synaptosomes from the VTA as shown by Western blotting, and in glutamatergic terminals of VTA-DA neurons as shown by the triple immunofluorescence staining method; and (3) selective stimulation of GABA neurons of the VTA with an optogenetic technique could inhibit the presynaptic glutamate release of VTA-DA neurons and the GABA$_B$ receptor antagonist CGP54626 could remove this inhibitory effect. These results strongly support the presence of an inhibitory circuit from neighboring GABA neurons to the presynaptic glutamatergic terminals of VTA-DA neurons. Moreover, they suggest that GABA$_B$ receptors mediate this inhibitory effect of GABA. On this basis, we checked whether morphine-promoted presynaptic glutamate release of VTA-DA neurons through the disinhibitory action on presynaptic glutamate release. To test this hypothesis, we used two strategies to remove inhibition by GABAergic input of presynaptic glutamate release and then observed their influence on the effect of morphine on glutamate release. The results showed that 'closing' local GABAergic interneurons both with optogenetic methods and with the GABA$_B$ receptor antagonist CGP54626 could abolish the promoting effect of morphine on glutamate release in VTA-DA neurons. These results confirm our hypothesis that morphine promotes presynaptic glutamate release in VTA-DA neurons through its disinhibitory action on presynaptic glutamate release.

It was previously reported that μ-opioid agonists hyperpolarized the GABAergic interneurons of the VTA, reduced the frequency of sIPSCs, and thus produced a disinhibitory action on VTA-DA

neurons (*Johnson and North, 1992*). In the present study, we demonstrated that morphine-disinhibited presynaptic glutamate release in VTA-DA neurons by a second mechanism. These two kinds of disinhibitory actions may both contribute to the morphine-induced excitatory effect on VTA-DA neurons. However, there is no direct evidence supporting the suggestion that the disinhibitory action on VTA-DA neurons causes the morphine-induced increase in DA neuron firing. In the present study, we investigated the contribution of the disinhibitory action of morphine on presynaptic glutamate release in VTA-DA neurons to the overall excitatory effect of morphine on VTA-DA neurons by employing the selective presynaptic GABA$_B$ receptor antagonist CGP36216. Several studies have indicated that pre- and postsynaptic GABA$_B$ receptors in central neurons may be pharmacologically distinct (*Dutar and Nicoll, 1988*; *Deisz et al., 1993*, *1997*). CGP36216 is a useful lead compound to differentiate between pre- and postsynaptic GABA$_B$ receptors. Up to 1 mM of CGP36216 was ineffective in antagonizing baclofen-induced hyperpolarization mediated through postsynaptic GABA$_B$ receptors, but the responses mediated through presynaptic GABA$_B$ receptors were reversibly antagonized by CGP36216 with an IC$_{50}$ of 43 µM (*Ong et al., 2001*). Therefore, in the present study, we evaluated the contribution of the disinhibitory action of morphine on presynaptic glutamate release in VTA-DA neurons to the overall excitatory effect of morphine on VTA-DA neurons by examining the effect of CGP36216 on the morphine-induced increase in VTA-DA neuron firing and related behaviors. Our results showed that if GABAergic inhibitory control of presynaptic glutamate release in VTA-DA neurons was removed, the effect of morphine on the firing of VTA-DA neurons disappeared, indicating that the morphine-induced increase in VTA-DA neuron activity might mainly depend on its disinhibitory action on presynaptic glutamate release. We also observe the effect of removing GABA inhibitory control of presynaptic glutamate release with the presynaptic GABA$_B$ receptor antagonist CGP36216 on morphine-induced increase in locomotor activity, which is often assessed as increased VTA-DA neuron activity at the behavioral level (*Vezina, 2004*). The results showed that intra-VTA injection of a presynaptic GABA$_B$ receptor antagonist could abolish the increase in rat locomotor activity induced by both i.p. and intra-VTA-injected morphine. In addition, we observed the effect of removing GABA inhibitory control of presynaptic glutamate release with the presynaptic GABA$_B$ receptor antagonist CGP36216 on morphine-induced CPP, which is initiated by morphine-induced activation of VTA-DA neurons (*Tsai et al., 2009*). The results showed that intra-VTA injection of CGP36216 could abolish morphine-induced CPP in rats. These data suggest that the disinhibitory action of morphine on presynaptic glutamate release may be the main mechanism in the morphine-induced increase in VTA-DA neuron firing and related behaviors.

## Materials and methods

### VTA slice preparation

Male Sprague–Dawley rats (14–16 days old) or mice (4–5 weeks old) were anesthetized with chloral hydrate (400 mg/kg, i.p.). All experimental procedures conformed to Fudan University as well as international guidelines on the ethical use of animals. All efforts were made to minimize animal suffering and reduce the number of animals used. VTA slices were prepared according to procedures described previously (*Hopf et al., 2007*). The brain was removed rapidly from the skull and placed in modified ACSF containing 75 mM sucrose, 88 mM NaCl, 2.5 mM KCl, 1.25 mM NaH$_2$PO$_4$, 7 mM MgCl$_2$, 0.5 mM CaCl$_2$, 25 mM NaHCO$_3$, and saturated with 95% O$_2$ and 5% CO$_2$ at ~0℃. Horizontal 250 µm midbrain slices containing VTA were cut on a vibratome (VT-1200, Leica, Wetzlar, Germany) and transferred to normal ACSF containing 126 mM NaCl, 2.5 mM KCl, 1.25 mM NaH$_2$PO$_4$, 2 mM MgSO$_4$, 2.5 mM CaCl$_2$, 25 mM NaHCO$_3$, and 10 mM glucose at 32℃. Slices were incubated for at least 60 min before patch-clamp recording.

### Whole-cell patch-clamp recording

The medial terminal nucleus of the accessory optic tract (MT) was used as the anatomical structure to define the VTA (*Hopf et al., 2007*). VTA neurons were visualized on an upright microscope (BX50WI, Olympus, Tokyo, Japan) using infrared differential interference contrast or fluorescent optics. Whole-cell current- and voltage-clamp recordings were made using an EPC10 amplifier and PatchMaster 2.54 software (HEKA, Lambrecht, Germany). Electrodes had a resistance of 3–4 MΩ when filled with the patch pipette solution. The internal pipette solution contained 130 mM K-gluconate, 8 mM NaCl, 0.1 mM CaCl$_2$, 0.6 mM EGTA, 2 mM Mg-ATP, 0.1 mM Na$_3$-GTP, and 10 mM HEPES (pH 7.4). Lucifer yellow

(2 mM) was added to the internal pipette solution for labeling the recorded neuron. Cells were held at 0 pA under a current-clamp mode to record spontaneous firing. Only neurons with regular spontaneous firing and an action potential amplitude greater than 60 mV were used for electrophysiological analyses. Cells were held at −70 mV under a voltage-clamp mode to record sEPSCs or evoked EPSC. A concentric stimulating electrode (FHC, Bowdoin, USA) was placed near the recorded cell. To observe PPF, two synaptic responses were evoked by a pair of stimulating pulses given at short intervals (50 ms) at 0.1 Hz. The series resistance (Rs) was monitored by measuring the instantaneous current in response to a 5 mV voltage step command. Rs compensation was not used, but cells where Rs changed by >15% were discarded.

## Identification of VTA-DA neurons

After forming whole-cell recording mode, we first identified DA neurons based on electrophysiological characteristics, which included both a spontaneous pacemaker-like firing and expression of a hyperpolarization-induced current ($I_h$) in the voltage-clamp configuration, by 1 s hyperpolarizing voltage steps (−70 mV to −150 mV) (*Grace and Onn, 1989*; *Margolis et al., 2006*; *Zhang et al., 2010*; *Chieng et al., 2011*) (*Figure 1A*). Next, DA neurons were retrospectively confirmed by labeling the recorded neuron with Lucifer yellow (2 mM) in the internal pipette solution and subsequent TH staining of the recorded cell with Lucifer yellow. Slices were fixed immediately after electrophysiological recording in 4% formaldehyde for 2 hr and then washed with 0.01 M PBS solution. The slices were incubated for 2 hr at 4°C in a blocking solution containing 10% normal goat serum and 0.2% Triton X-100 in PBS and then incubated overnight at 4°C with primary rabbit anti-TH antibody (1:1000; Abcam, Cambridge, UK). The slices were washed thoroughly in PBS before being incubated for 2 hr at 4°C with goat anti-rabbit-Cy3 secondary antibody (1:200; Jackson ImmunoResearch Laboratories, West Grove, USA). The recorded cells were identified as DA neurons based on the co-labeling by Lucifer yellow and TH (*Figure 1B*).

## Synaptosome preparation

Male Sprague–Dawley rats (100–150 g) were anesthetized with chloral hydrate (400 mg/kg, i.p.). Synaptosomes were prepared as described previously (*Dong et al., 2005*). The VTA was dissected and homogenized in 0.32 M sucrose solution at 4°C using the Art-Miccra D-8 tissue grinder with a motor-driven pestle rotating at 900 rpm. The homogenate was centrifuged at 3000×$g$ for 3 min at 4°C. The supernatant (S1) was centrifuged at 14,500×$g$ for 12 min at 4°C. The pellet (P2) was resuspended and loaded onto Percoll gradients consisting three steps of 23%, 10%, and 3% Percoll in 0.32 M sucrose additionally containing 1 mM EDTA and 250 μM DTT. The gradients were centrifuged at 32,500×$g$ for 6.5 min at 4°C. Synaptosomes were harvested from the interface between the 23% and 10% Percoll layers and washed in Hanks' balanced salt solution (HBSS) containing 140 mM NaCl, 5 mM KCl, 5 mM NaHCO₃, 1 mM MgCl₂, 1.2 mM Na₂HPO₄, 10 mM glucose, and 20 mM HEPES, pH 7.4. Washed synaptosomes were centrifuged at 27,000×$g$ for 15 min at 4°C.

## Measurement of glutamate release from synaptosomes

Glutamate release from synaptosomes was assayed by on-line fluorimetry as described previously (*Nicholls et al., 1987*). Synaptosomal pellets were suspended again by adding 1.5 ml of incubation medium: 122 mM NaCl, 3.1 mM KCl, 0.4 mM KH₂PO₄, 5 mM NaHCO₃, 20 mM Na-TES, 1.2 mM MgSO₄, 16 pM bovine serum albumin (fatty acid free, type V), and 10 mM D-glucose, pH 7.4. Additionally, NADP⁺ (2 mM), glutamate dehydrogenase (50 units/ml), and CaCl₂ (1 mM) were added after 3 min. After a 5-min incubation period, morphine (10 μM) was added to observe its effect on glutamate release. Oxidative decarboxylation of released glutamate, leading to reduction in NADP⁺, was monitored by measuring NADPH fluorescence at excitation and emission wavelengths of 340 nm and 460 nm, respectively. Data points were obtained at 60 s intervals. An exogenous glutamate standard (5 nmol) was added at the end of each experiment. The fluorescence change produced by the addition of exogenous glutamate (5 nmol) was used to calculate the released glutamate in nmol/mg.

## Mechanical dissociation of VTA-DA neurons

VTA-DA neurons with functional presynaptic terminals attached were obtained by a mechanical dissociation method as previously described (*Akaike and Moorhouse, 2003*; *Ye et al., 2004*;

*Deng et al., 2009*). First, VTA slices were prepared as described above. Then the slice was transferred to a 35-mm culture dish (Sigma, USA) and held down with a flat U-shaped wire. The dish was filled with standard external solution containing 140 mM NaCl, 5 mM KCl, 2 mM $CaCl_2$, 1 mM $MgCl_2$, 10 mM HEPES, and 10 mM glucose (pH 7.4). The MT was used as the anatomical structure to define the VTA (*Hopf et al., 2007*) under a stereomicroscope (SZ61, Olympus). A fire-polished glass pipette, lightly touching the surface of the VTA, was vibrated horizontally at 50–60 Hz for 1–2 min by a home-made device. The slice was then removed. The isolated neurons adhered to the bottom of the dish within 20 min and were then ready for electrophysiological experiments.

## Western blotting

Immunoblot analysis of $GABA_B$ R1 receptors was performed on the synaptosomes obtained from the VTA. The synaptosomal pellets were homogenized in a buffer containing 100 mM Tris-HCl (pH 6.7), 1% SDS, 143 mM 2-mercaptoethanol, and 1% protease inhibitor. The lysate was centrifuged at 12,000 rpm for 10 min at 4°C. The samples were treated with the SDS sample buffer at 95°C for 5 min, loaded on a 10% SDS polyacrylamide gel, and blotted to a PVDF membrane. Each blot was incubated with a rabbit anti-$GABA_B$ R1 (1:100; Alomone Labs, Jerusalem, Israel). Following extensive washing, membranes were incubated with IRDye 800CW goat anti-rabbit secondary antibodies (1:20,000; LI-COR, Lincoln, USA) for 1 hr, and images were acquired on a LI-COR Odyssey system.

## Immunohistochemistry

Sprague–Dawley rats (15–20 days old) were perfused transcardially with 0.01 M PBS followed by 4% paraformaldehyde. After perfusion, brains were removed and postfixed in 4% paraformaldehyde overnight. Horizontal brain slices (40-μm thick) containing VTA were prepared using a vibratome (VT-1000S, Leica). Slices were pre-blocked for 2 hr at 4°C in a blocking solution containing 10% horse serum and 0.2% Triton X-100 in PBS and again washed three times in PBS for 5 min. Slices were incubated overnight at 4°C with primary antibody mouse anti-TH (1:1000; Millipore, Billerica, USA), rabbit anti-$GABA_B$ R1 (1:100; Alomone Labs), and guinea pig anti-VGLUT2 (1:3000; Millipore) dissolved in the blocking solution. Afterwards, slices were washed three times in PBS for 5 min and then incubated with the following secondary antibodies: horse anti-rabbit-Alexa 488 (1:200; Jackson ImmunoResearch Laboratories), horse anti-mouse-Cy5 (1:200; Jackson ImmunoResearch Laboratories), and horse anti-guinea pig-Cy3 (1:200; Jackson ImmunoResearch Laboratories) for 2 hr at room temperature in 2% horse serum and 0.2% Triton X-100 in PBS. Subsequently, slices were washed three times in PBS for 5 min, and sections were mounted on glass slides using aqua-mount mounting medium (Thermo Fisher Scientific, Waltham, USA). Confocal images of sections were obtained using confocal microscopy (FV1000, Olympus) with a 60× oil-immersion lens.

## In vitro optogenetic approach for electrophysiology

The optogenetic approach was based on procedures described previously (*van Zessen et al., 2012*). Bilateral injections of purified and concentrated pAAV-EF1a-double floxed-hChR2 (H134R)-mCherry or pAAV-EF1a-double floxed-eNpHR 3.0-EYFP virus ($2.05 \times 10^{12}$ vector genomes/ml; Neuron Biotech Company, Shanghai, China) were stereotaxically performed in 3-week-old male Gad2-IRES-Cre mice (B6N.Cg-Gad2$^{tm2(cre)}$Zjh/J; Jackson Laboratories, USA). Each side of the VTA (final coordinates: AP, −3.1 mm; ML, ±0.4 mm; DV, −4.2 mm from the skull surface) was injected with 0.5 μl AAV for 10 min followed by an additional 10 min to allow diffusion of viral particles away from the injection site. About 10–14 days after AAV virus injection, VTA slices were prepared according to the procedures described above. GABA neurons were stimulated by a 5 ms, 470 nm light or a constant 590 nm light delivered via an optical fiber (core diameter 200 μm, NA 0.39; ThorLabs, USA) coupled to an LED light source (Mightex, California, USA). The end of the fiber optic cannula (stainless ferrule, core diameter 200 μm, length 20 mm; ThorLabs, New Jersey, USA) connected to the optical fiber through a mating sleeve (ThorLabs) was placed 500 μm above the recording cell.

## Surgery

Male Sprague–Dawley rats (270–300 g) were used. Animals were housed singly under a 12 hr light/dark cycle (lights on 7:00 AM to 7:00 PM) in a temperature- and humidity-controlled environment with food and water freely available. The rats were anesthetized with chloral hydrate (400 mg/kg, i.p.) and

placed in stereotaxic instruments (Stoelting, Wood Dale, USA). Two 24-gauge stainless steel guide cannulae were implanted bilaterally 2 mm above the VTA. The coordinates were: AP, −5.8 mm; ML, ±2.5 mm; DV, −8.0 mm from the skull surface with a 14˚ lateral angle. The cannulae were secured to the skull with two anchoring screws and dental cement. To prevent occlusion, 30-gauge wire plugs were inserted into the cannulae. After the surgery, the animals were housed individually and were allowed to recover for over a week.

## Locomotor behavior

The locomotor activity test was conducted as described previously with some modifications (*Borgland et al., 2006*). The locomotor activity of animals was monitored with a near infrared video camera within the operant chambers (Med Associates, St. Albans, USA). Distance traveled was measured using Open Field Activity Software (Med Associates) and analyzed locomotion estimates based on movement over a given distance and resting delays (movement in a given period of time). All animals were habituated to the test room for 2 hr prior to the start of the experiment. Rats were further habituated to the operant chambers for 30 min prior to the 120 min testing session. For bilateral intra-VTA microinjection, 30-gauge injection needles were inserted into the cannulae. The injection needles were connected to a 1-μl microsyringe (Hamilton, Reno, USA) by a polyethylene tube and controlled by a syringe pump (Harvard Apparatus, Holliston, USA). After injection, the needles were left in place for another 1 min. Rats were bilaterally injected with 0.2 μl morphine (0.5 μg/side) or CGP36216 (10 μg/side) for 1 min. After drug administration, the rats were placed in the chambers for the 120 min testing session. On days 1, 2, and 3, all rats were only given saline (bilateral intra-VTA microinjections or intraperitoneal injections) to habituate them to the test protocol. On day 4, rats were given drugs according to experimental group. After the behavioral tests, all rats were anesthetized with an overdose of chloral hydrate and perfused with 0.9% saline. The brain was removed and fixed in 4% paraformaldehyde for 24 hr. Coronal sections (80 μm) were cut by a vibratome and stained with cresyl violet. Injection sites were verified under light microscope. Animals where the injection site was outside the VTA were discarded.

## Conditioned place preference

The CPP test was conducted using a three-compartment place conditioning apparatus (Med Associates) with distinct tactile environments to maximize contextual differences. The procedure for CPP testing was similar to that described previously (*Phillips and LePiane, 1980*; *Bardo and Neisewander, 1986*; *Wang et al., 2008*), with some modifications. On day 1, rats underwent a preconditioning test: they were placed in the middle neutral area and were allowed to freely access both sides of the apparatus for 15 min. Rats with a strong preference (60%) for any compartment were discarded. Conditioning was performed using an unbiased, balanced protocol. On day 2, rats were microinjected with saline, morphine, CGP36216, or CGP36216 plus morphine 30 min before they were confined to the conditioning chambers for 45 min (drug-paired). On day 3, rats received a microinjection of saline and were then confined to the other chamber for 45 min. The day after conditioning, rats were tested for drug-induced (postconditioning test) CPP in the same conditions as for the preconditioning test. The place preference score (CPP score) was defined as the time spent in the drug-paired chamber minus that spent in the drug-unpaired (saline-paired) chamber. Injection sites were verified under light microscope as described above.

## Drugs

Methyl sulfoxide (DMSO), PTX, K-gluconate, Lucifer yellow, $K_2$-ATP, $Na_3$-GTP, 4-(2-hydroxyethyl) piperazine-1-ethanesulfonic acid (HEPES), N-[tris(hydroxymethyl)methyl]-2-aminoethanesulfonic acid (Na-TES), nicotinamide adenine dinucleotide phosphate ($NADP^+$), glutamate dehydrogenase, sodium dodecyl sulfate (SDS), ethyleneglycol-bis(β-aminoethyl ether)-N,N,N′,N′-tetraacetic acid (EGTA), Triton X-100, and 0.01 M PBS were purchased from Sigma. [S-(R*,R*)]-[3-[[1-(3,4-Dichlorophenyl) ethyl]amino]-2-hydroxypropyl] (cyclohexylmet-hyl) phosphinic acid (CGP54626) and (3-aminopropyl) ethylphosphinic acid hydrochloride (CGP36216) were purchased from Tocris. Morphine was from Shenyang No. 1 Pharmaceutical Factory, China. Percoll was purchased from Amersham Biosciences Corporation. Other AR grade reagents were from Shanghai Chemical Plant. PTX or CGP54626 was dissolved in DMSO and others were dissolved in ddH$_2$O. When DMSO was used as the vehicle, drugs

were initially dissolved in 100% DMSO and then diluted into ASFC at a final DMSO concentration of less than 0.5%, which had no detectable effects on the parameters we observed.

## Data analysis

Numerical data were expressed as the mean ±s.e.m. Off-line data analysis was performed using the Mini Analysis Program (Synaptosoft, Fort Lee, USA), Clampfit (Axon Instruments, Sunnyvale, USA), SigmaPlot (Systat Software, San Jose, USA), and Origin (Microcal Software, Northampton, USA). Spontaneous firing was analyzed using the Event Detection function of Clampfit. sEPSCs were analyzed using the Mini Analysis Program. Detection criteria were set at >8 pA, <1 ms rise-time, and <3 ms decay-time for sEPSCs (*Borgland et al., 2006*). Statistical significance was determined using Student's *t*-test for comparisons between two groups or ANOVAs followed by the Student–Newman–Keuls test for comparisons among three or more groups. In the patch-clamp studies, n refers to the number of cells. Every cell was from a different slice, and a group of cells in each experiment was from at least four animals. Two-way ANOVAs were performed on the data from CPP with the between-subjects factors treatment (different drugs). All post-hoc comparisons were made using Tukey's test. Results with p < 0.05 were accepted as being statistically significant.

## Acknowledgements

We would like to thank Dr Jun Chen and Dr Ruth Anne Stetler at the University of Pittsburgh for critically reading this manuscript and offering many suggestions.

---

## Additional information

### Funding

| Funder | Grant reference | Author |
| --- | --- | --- |
| National Basic Research Program of China | 2015CB553500 | Ping Zheng |
| Science and Technology Program of Yunnan Province | 2013GA003 | Ping Zheng |
| National Basic Research Program of China | 2013CB835100 | Ping Zheng |
| National Basic Research Program of China | 2009CB52201 | Ping Zheng |
| National Natural Science Foundation of China | 31121061 | Ping Zheng |
| National Natural Science Foundation of China | 91332204 | Ping Zheng |
| National Natural Science Foundation of China | 81371466 | Ping Zheng |
| National Natural Science Foundation of China | 31070932 | Ping Zheng |
| National Natural Science Foundation of China | 31421091 | Ping Zheng |

The funders had no role in study design, data collection and interpretation, or the decision to submit the work for publication.

---

### Author contributions

MC, Conception and design, Acquisition of data, Analysis and interpretation of data, Drafting or revising the article; YZ, HY, WL, JS, DC, YD, BL, Acquisition of data, Analysis and interpretation of data; LM, Conception and design, Analysis and interpretation of data; PZ, Conception and design, Analysis and interpretation of data, Drafting or revising the article

### Ethics

Animal experimentation: All experimental procedures conformed to Fudan University as well as international guidelines on the ethical use of animals and all efforts were made to minimize the number of animals used and their suffering.

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
