## [Decision Letter]

Thank you for submitting your work entitled “Morphine disinhibits glutamatergic input onto VTA dopamine neurons and promotes dopamine neuron excitation” for further consideration at *eLife*. Your new submission has been evaluated by a Senior Editor, a Reviewing Editor, and two of the original reviewers.

The manuscript has been improved scientifically and would be acceptable after revision of the Abstract and indicating the species for all data in the Results section and in the legends. Please address the following concerns raised by Reviewer 2.

*Reviewer #2*:

The authors have made many changes, including addition of new data. These strengthen the conclusions, and my original queries have been addressed. However I can't help feeling that further changes are needed for clarity. A few examples:

Abstract:

“The model for morphine to activate VTA-DA neurons is VTA-DA neuron disinhibition model.”

“…the main mechanism that leads to morphine-induced increase in VTA-DA neurons firing…”

Introduction:

“This inhibitory effect on glutamate release has being confusing (8).”

“Moreover, the strengthening effect of morphine on glutamatergic input onto VTA-DA neurons is more relevant to its promoting effect on VTA-DA neurons excitation.”

“…using whole-cell patch-clamp method…”

Two other important but easily fixable points:

1) I think the Abstract does not convey the level of detail to which the mechanism of the effect of morphine has been explored. Instead it almost implies that results (1), (2) and (3) are separate, unrelated observations.

2) The Results text and the figure legends (and as far as possible the Abstract) need to be clear on which data are from which species.

[Editors’ note: a previous version of this study was rejected after peer review, but the authors submitted for reconsideration. The previous decision letter after peer review is shown below.]

Thank you for choosing to send your work entitled “Morphine disinhibits glutamatergic inputs onto VTA dopamine neurons and promotes dopamine neuron excitation” for consideration at *eLife*. Your full submission has been evaluated by a Senior Editor and three peer reviewers, one of whom is a member of our Board of Reviewing Editors, and the decision was reached after discussions between the reviewers.

Based on our discussions and the individual reviews below, we regret to inform you that your work in its current state will not be considered for publication in *eLife*. However, there was a consensus among the reviewers and editors that your work is very interesting, and that we would be keen to publish it, but only if you were able to satisfactorily address all reviewers' issues, including the additional experiments suggested.

During the reviewing process, the following issues came up that must be addressed in a new submission, together with all points made by the three reviewers:

1) Additional experiments are required on:

A) Confirmation that authors were actually measuring sEPSCs: do recordings of sEPSCs as suggested by referee.

B) Repeat some of the key experiments side-by side in both spices.

C) CPP test in order to observe the influence of CGP36216 on the effect of morphine reward.

D) Add a control with the GABA antagonist alone as required by the 2x2 factorial design.

2) Address issues with the figures:

A) Cite more Figure 1 and simplify Figure 7.

B) Some documentation is missing, as detailed in the reviewers' comment, which would require additional figures.

C) Show all Western blots.

3) Description and interpretational issues, as detailed in the reviewers' reports.

Reviewer #1:

The authors have carried out a comprehensive set of experiments and present some convincing data to support the idea that acute morphine increases the activity of VTA dopamine neurons via disinhibition of presynaptic glutamate release. Using a range of clever approaches, small but significant increases in spontaneous synaptic event frequency are seen with morphine that appears to depend on the presence of local inhibitory neurons in the VTA and GABA(B)R activity.

Main concerns:

1) sEPSCs are usually recorded when sIPSCs are blocked with picrotoxin (PTX) or bicuculline in the bath solution. This was not routinely used in the manuscript, as PTX in the bath causes morphine to inhibit rather than increase glutamate release. It is important to show that the spontaneous events that the authors measure (control and after morphine) are in fact sEPSCs, i.e. that they are abolished by glutamate receptor antagonists.

2) Figure 7: What is the purpose of Figure 7? Why are the effects of PTX and CGP alone suddenly investigated at this stage? It's helpful to know that PTX does not block the spontaneous events (see above), but this does not seem the right place to show it. Figure 7 is large, and the description of it in the text is lengthy, so it should be simplified as far as possible.

3) [9] report that glutamate release is decreased in Mu receptor KO mice, suggesting that Mu receptors ordinarily increase glutamate release. Chefer et al. should be discussed in this manuscript.

Other points:

1) Figure 2 and Figure 9: The y-axes are truncated, which is misleading.

2) In the subsection “Morphine has an additional promoting effect on presynaptic glutamate release in VTA-DA neurons”: What do the authors mean by ‘glutamatergic tone’? The data suggest that the effect of morphine requires AMPARs and/or NMDARs (although the data do not show that the AMPARs and NMDARs required are necessarily on dopamine neurons), not that the effect of morphine depends on glutamatergic ‘tone’.

3) In Figure 2 the firing rate in the presence of glutamate receptor antagonists looks faster than in control (Figure 2). Could the authors comment on this.

4) Figure 1 is not necessary.

5) Introduction, first paragraph: “However, the mechanism behind activation of VTA-DA neurons by morphine remains to be studied.” This statement should be deleted or at least re-phrased as the mechanism of morphine-induced disinhibition of VTA DA neurons is very well studied, even if the authors have uncovered some new details.

Reviewer #2:

This manuscript entitled “Morphine disinhibits glutamatergic inputs onto VTA dopamine neurons and promotes dopamine neuron excitation” examined the mechanism by which morphine affects the firing of dopamine neurons in the ventral tegmental area (VTA). Previous studies have suggested that morphine inhibits GABAergic neurons in the VTA (or the rostromedial tegmental area [RMTg]) to disinhibit dopamine neurons. In the present study, the authors performed a series of experiments in slice and behavior to indicate that morphine increases the frequency of glutamate release onto dopamine neurons. The authors first showed that the frequency of spontaneous excitatory postosynaptic currents (sEPSCs) increased after morphine application in slice. To isolate sEPSCs, the authors blocked inhibitory currents in the recorded neuron by applying a GABA-A receptor antagonist, picrotoxin, from the internal solution of the patch pipette (2). The authors showed increasing VTA GABAergic neuron firing decreased sEPSCs while blocking GABA-B receptors increased sEPSCs. Furthermore, the effect of morphine to increase locomotor activity was blocked by local injection of GABA-B receptor agonist in the VTA.

This study provides a novel idea regarding the action of morphine on dopamine neurons, which is potentially of great significance. However, there were a couple points that were not convincing. I would like to see the authors' response.

Major issues:

1) The authors recorded sEPSCs while VTA GABAergic neurons were activated by optogenetically. I would expect that VTA GABAergic neurons provide strong direct inhibitions onto dopamine neurons. Why did the authors not see this effect? Did the authors use intracellular picrotoxin in this experiment as well? It was not stated in the manuscript.

2) In Figure 10, the authors apply GABA-B antagonist, CGP36216, to counteract morphine's effects on locomotor activity. The authors show that applying CGP36216 reduced the effect of morphine. However, the authors' results suggest that both morphine, applied without GABA-A blocker, and CGP3616 increase dopamine neuron firing. It was unclear how the authors interpret these results mechanistically.

Reviewer #3:

It has been shown that the rewarding effects of opioids occur through activation of VTA DA neurons but the molecular mechanism underlying this activation are still debated, especially the effect of morphine on the glutamatergic transmission, which is the focus of the current paper. Using a multidisciplinary approach, combining electrophysiology, optogenetics, biochemistry and behavior, Zheng et al. investigated the effect of morphine on glutamatergic inputs onto VTA DA neurons. In addition to the accepted disinhibition model, Zheng et al. argue for an excitatory effect of morphine on presynaptic glutamate release.

I would have some comments in order to strengthen the paper.

1) A major issue with this paper is the use of mice for the optogenetic experiments and rats for the rest. Rats and mice are species showing significant differences in terms of behavior and neurobiology. In order to be able to make comparisons and generalizations I would use the same species for all of the study, or at least, the authors must repeat some of the key experiments side-by side in both spices.

2) The aims of the study are well defined and the background is solid however I would add the following references. Concerning the disinhibition model, in addition to the Johnson and North paper I would add these two papers:

Kalivas PW (1993) Neurotransmitter regulation of dopamine neurons in the ventral tegmental area. Brain Res Rev 18:75-113.

White FJ (1996) Synaptic regulation of mesocorticolimbic dopamine neurons. Annu Rev Neurosci 19:405-436.

In the same line, I suggest including the study of Bonci, A. and Malenka, R.C. (1999) Properties and plasticity of excitatory synapses on dopaminergic and GABAergic cells in the ventral tegmental area. J. Neurosci. 19, 3723-3730, when speaking about the inhibition of the glutamate terminals onto VTA DA neurons.

3) There is no description of Figure 1 in the text. The identification of DA neurons by the presence of an I_h_ current is not ideal and can be misleading. Did the authors consider using genetic methods for the identification?

4) In Figure 2 the authors should explain the choice of the morphine dose. Does it rely on a previous study, is it a physiological dose?

5) Where is the figure showing that morphine has no significant effect on the amplitude of sEPSCs? Likewise the figure showing that the morphine has no effect on glutamate release from synaptosomes of the VTA is missing. Also missing is the figure showing that the CGP36216 could increase the firing rate of VTA-DA neurons.

6) In Figure 2, morphine had a strong excitatory effect on VTA DA neurons, which is no longer present on the dissociated DA neurons used in Figure 4. The authors must refer to the results of Figure 2 in the Figure 4 description and detail the hypothesis underlying such difference.

7) For Figure 6, the authors should show all western blots (including the control) and not just a small part of it. Likewise, I would describe what chemical was used to label the GABA(B)R.

8) Morphine has been shown to induce CPP. In order to strengthen the behavioral part, the authors should perform a CPP test in order to observe the influence of CGP36216 on the effect of morphine reward. All four groups of the 2x2 factorial design must be reported.

---

## [Author Response]

The manuscript has been improved scientifically and would be acceptable after revision of the Abstract and indicating the species for all data in the Results section and in the legends. Please address the following concerns raised by Reviewer 2.

In the present revised version, we revised the Abstract and indicated the species for all data in the Results section and in the legends. We also revised the examples pointed out by Reviewer 2.

Reviewer #2:

The authors have made many changes, including addition of new data. These strengthen the conclusions, and my original queries have been addressed. However I can't help feeling that further changes are needed for clarity. A few examples.

Abstract:

“The model for morphine to activate VTA-DA neurons is VTA-DA neuron disinhibition model.”

Now we have revised it as “One reported mechanism for morphine activation of dopamine (DA) neurons of the ventral tegmental area (VTA) is the disinhibition model of VTA-DA neurons.”

*“*…*the main mechanism that leads to morphine-induced increase in VTA-DA neurons firing*…*”*

Now we have revised it as “…the main mechanism for morphine-induced increase in VTA-DA neuron firing…”.

Introduction:

*“This inhibitory effect on glutamate release has being confusing (*[8]*).”*

Now we have revised it as “This inhibitory effect of opioids on glutamate release is puzzling (8)”.

“Moreover, the strengthening effect of morphine on glutamatergic input onto VTA-DA neurons is more relevant to its promoting effect on VTA-DA neurons excitation.”

Now we have revised it as “Moreover, the strengthening influence of morphine on glutamatergic input to VTA-DA neurons is more relevant to its promoting effect on VTA-DA neuron excitation.”

*“*…*using whole-cell patch-clamp method*…*”*

Now we have revised it as “using the whole-cell patch-clamp method…”.

Two other important but easily fixable points:

1) I think the Abstract does not convey the level of detail to which the mechanism of the effect of morphine has been explored. Instead it almost implies that results (1), (2) and (3) are separate, unrelated observations.

We have edited the Abstract.

2) The Results text and the figure legends (and as far as possible the Abstract) need to be clear on which data are from which species.

In the present revised version, we indicated the species for all data in the Results section and in the legends.

[Editors’ note: the author responses to the previous round of peer review follow.]

During the reviewing process, the following issues came up that must be addressed in a new submission, together with all points made by the three reviewers:

1) Additional experiments are required on:

A) Confirmation that authors were actually measuring sEPSCs: do recordings of sEPSCs as suggested by referee.

We examined the influence of AMPA receptor antagonist DNQX (10 μM) on spontaneous events that we measured under control and after morphine in the condition of intracellularly applied PTX. We added these results into the present revised version as follows:

“In addition, to further confirm that the spontaneous events we measured in the presence of intracellularly applied PTX were in fact sEPSCs, we observed the influence of AMPA receptor antagonist DNQX on spontaneous events in the presence of intracellularly applied PTX. The result showed that the spontaneous events were completely blocked by adding AMPA receptor antagonist DNQX (10 μM) (bottom of panel 1 in Figure 3). We repeated the experiment in 3 cells from different slices and obtained a similar result.”

“To confirm the increased spontaneous events we measured in the presence of intracellularly applied PTX after morphine were also in fact sEPSCs, we observed the influence of AMPA receptor antagonist DNQX on spontaneous events after morphine in the presence of intracellularly applied PTX. The result showed that the spontaneous events after morphine were completely blocked by adding AMPA receptor antagonist DNQX (10 μM) (panel 6 of Figure 3). We repeated the experiment in 3 cells from different slices and obtained a similar result.”

B) Repeat some of the key experiments side-by side in both spices.

Based on the reviewer’s suggestion, we repeat some key experiments in mice. Please see below.

C) CPP test in order to observe the influence of CGP36216 on the effect of morphine reward.

We added this experiment in the present revised version.

D) Add a control with the GABA antagonist alone as required by the 2x2 factorial design.

We added this control.

2) Address issues with the figures:

*A) Cite more*
Figure 1
*and simplify*
Figure 7*.*

More references to Figure 1 were added, and we simplified Figure 7 in the present revised version.

B) Some documentation is missing, as detailed in the reviewers' comment, which would require additional figures.

We added these additional figures based on the reviewer’s suggestion.

C) Show all Western blots.

In the present revised version, we showed all Western blots.

3) Description and interpretational issues, as detailed in the reviewers' reports.

We added text involving the description and interpretational issues by the reviewers.

Reviewer #1:

Main concerns:

1) sEPSCs are usually recorded when sIPSCs are blocked with picrotoxin (PTX) or bicuculline in the bath solution. This was not routinely used in the manuscript, as PTX in the bath causes morphine to inhibit rather than increase glutamate release. It is important to show that the spontaneous events that the authors measure (control and after morphine) are in fact sEPSCs, i.e. that they are abolished by glutamate receptor antagonists.

We examined the influence of AMPA receptor antagonist DNQX (10 μM) on spontaneous events that we measured under control and after morphine in the condition of intracellularly applied PTX. We added these results into the present revised version as follows:

“In addition, to further confirm […] obtained a similar result.”

“To confirm the increased spontaneous events […] 3 cells from different slices and obtained a similar result.”

*2)*
Figure 7*: What is the purpose of*
Figure 7*? Why are the effects of PTX and CGP alone suddenly investigated at this stage? It's helpful to know that PTX does not block the spontaneous events (see above), but this does not seem the right place to show it.*
Figure 7
*is large, and the description of it in the text is lengthy, so it should be simplified as far as possible.*

We agree with the reviewer’s comment about Figure 7 and the description of it in the text. In the present revised version, we reorganized Figure 5, Figure 7 and Figure 8 into three parts to address three questions: 1. Can GABA, including exogenous and intrinsic GABA, inhibit presynaptic glutamate release in VTA-DA neurons? (Figure 5); 2. How does GABA inhibit presynaptic glutamate release in VTA-DA neurons? (Figure 6); 3. Is there a basal GABAergic inhibitory control by GABA_B_ receptors on presynaptic glutamate release in VTA-DA neurons? (Figure 8).

*3)*
[9]
*report that glutamate release is decreased in Mu receptor KO mice, suggesting that Mu receptors ordinarily increase glutamate release. Chefer et al. should be discussed in this manuscript.*

In the present revised version, we added a discussion about this work (please see the passage: “Another possible mechanism […] via the inhibition of GABAergic interneurons in VTA-DA neurons.”)

Other points:

*1)*
Figure 2
*and*
Figure 9*: The y-axes are truncated, which is misleading.*

We redraw these figures with full y-axes in the present revised version.

2) In the subsection “Morphine has an additional promoting effect on presynaptic glutamate release in VTA-DA neurons”: What do the authors mean by ‘glutamatergic tone’? The data suggest that the effect of morphine requires AMPARs and/or NMDARs (although the data do not show that the AMPARs and NMDARs required are necessarily on dopamine neurons), not that the effect of morphine depends on glutamatergic ‘tone’.

We revised this statement in the present revised version as follows:

“These results suggest that the morphine-induced increase in spontaneous firing frequency of VTA-DA neurons requires AMPA and NMDA receptor-mediated glutamatergic input, being consistent with a recent report using the NMDA antagonists APV and AMPA receptor antagonist CNQX in in vivo experiment (24).”

*3) In*
Figure 2
*the firing rate in the presence of glutamate receptor antagonists looks faster than in control (*Figure 2*). Could the authors comment on this.*

The cell of Figure 2 is a different cell from that in Figure 2. Figure 2 showed firing traces under control and morphine from one cell. Figure 2 showed firing traces under control (DNQX+APV) and morphine (DNQX+APV) from another cell. So the control in Figure 2 is not the control of morphine (DNQX+APV) of Figure 2.

*4)*
Figure 1
*is not necessary.*

In the present revised version, we cite Figure 1 more, as suggested by Reviewer 2 and the Reviewing Editor.

5) Introduction, first paragraph: “However, the mechanism behind activation of VTA-DA neurons by morphine remains to be studied.” This statement should be deleted or at least re-phrased as the mechanism of morphine-induced disinhibition of VTA DA neurons is very well studied, even if the authors have uncovered some new details.

We deleted this statement in the revised version.

Reviewer #2:

[…] This study provides a novel idea regarding the action of morphine on dopamine neurons, which is potentially of great significance. However, there were a couple points that were not convincing. I would like to see the authors' response.

Major issues:

1) The authors recorded sEPSCs while VTA GABAergic neurons were activated by optogenetically. I would expect that VTA GABAergic neurons provide strong direct inhibitions onto dopamine neurons. Why did the authors not see this effect? Did the authors use intracellular picrotoxin in this experiment as well? It was not stated in the manuscript.

Yes, we used intracellular picrotoxin in this experiment as well. The intracellular picrotoxin abolished the inhibitory effect of the light-induced activation of GABAergic neurons on dopamine neurons, but it did not affect the inhibitory effect of the light-induced activation of GABAergic neurons on presynaptic glutamate release of VTA-DA neurons. We added a statement about it in the present revised version as follows:

“On this basis, whole-cell patch-clamp recording was performed in VTA-DA neurons to observe the effect of 470 nm light stimulation on the frequency of sEPSCs in the presence of intracellularly applied PTX (100 µM).”

*2) In*
Figure 10*, the authors apply GABA-B antagonist, CGP36216, to counteract morphine's effects on locomotor activity. The authors show that applying CGP36216 reduced the effect of morphine. However, the authors' results suggest that both morphine, applied without GABA-A blocker, and CGP3616 increase dopamine neuron firing. It was unclear how the authors interpret these results mechanistically.*

Our interpretation is as follows:

Figure 10 showed that if using the presynaptic GABA_B_ antagonist CGP36216 to remove the inhibition of GABAergic tone on presynaptic glutamate release before morphine, the effect of morphine on locomotor activity was counteracted. This result suggests that the effect of morphine depends on the disinhibitory action on presynaptic glutamate release.

The results that both morphine, applied without GABA_A_ antagonist, and CGP36216 increased dopamine neuron firing were in consistent with the above behavioral experiments. Our results showed that morphine increased DA-neuron firing via removing GABA_B_ receptor-mediated inhibition of presynaptic glutamate release, whereas the GABA_B_ receptor antagonist CGP36216 mimicked the increasing effect of morphine on firing via antagonizing presynaptic GABA_B_ receptors and removing GABA_B_ receptor-mediated inhibition of presynaptic glutamate release.

Reviewer #3:

[…] I would have some comments in order to strengthen the paper.

1) A major issue with this paper is the use of mice for the optogenetic experiments and rats for the rest. Rats and mice are species showing significant differences in terms of behavior and neurobiology. In order to be able to make comparisons and generalizations I would use the same species for all of the study, or at least, the authors must repeat some of the key experiments side-by side in both spices.

Based on the suggestion by the reviewer, we repeated some key experiments in mice as follows:

A) Morphine increases the frequency of sEPSCs in VTA-DA neurons of mice.

From raw sEPSC traces of DA neurons (Figure 3–panel 3) and time course of sEPSCs (Figure 3), we found that morphine (10 μM) increased the frequency of sEPSCs. The average frequency of sEPSCs increased from 3.2 ± 0.2 Hz before to 4.0 ± 0.2 Hz during 10-15 min after addition of morphine (n = 6, paired *t* test, *P* < 0.05, compared to control before morphine, Figure 3).

B) Exogenous application of GABA inhibits the frequency of sEPSCs in VTA-DA neurons of mice and GABA_B_ receptor antagonist CGP54626 abolishes the effect of GABA.

From raw current traces (left panel of Figure 13) and time course of sEPSCs (middle panel of Figure 13), we could see that GABA (10 μM) apparently decreased the frequency of sEPSCs. The average frequency of sEPSCs decreased from 3.4 ± 0.2 Hz before to 2.1 ± 0.2 Hz during 10-15 min after GABA (n = 6, paired *t* test, *P* < 0.05, compared to control before GABA, right panel of Figure 13). In the presence of GABA_B_ receptor antagonist CGP54626, the effect of GABA on the frequency of sEPSCs disappeared (Figure 13). The average frequency of sEPSCs was 3.6 ± 0.4 Hz before to 3.4 ± 0.4 Hz during 10-15 min after GABA in the presence of CGP54626 (2 µM) (n = 6, paired *t* test, *P* > 0.05, compared to CGP54626 before GABA, right panel of Figure 13).

Author response image 1.Effect of GABA on the frequency of sEPSCs and the influence of GABA_B_ receptor antagonist CGP54626 on the effect of GABA on the frequency of sEPSCs in VTA-DA neurons of mice.(**A**) Effect of GABA on the frequency of sEPSCs in VTA-DA neurons. Left panel: Typical current traces of sEPSC before and after GABA (10 μM). Middle panel: Time course of the frequency of sEPSCs before and after GABA (10 μM). Right panel: Average frequency of sEPSCs before and after GABA (10 μM) (*n* = 6, *P* < 0.05, compared to control before GABA). (**B**) Influence of GABA_B_ receptor antagonist CGP54626 on the effect of GABA. Left panel: Typical current traces of sEPSC before and after GABA (10 μM) in the presence of CGP54626 (2 μM). Middle panel: Time course of the frequency of sEPSCs before and after GABA (10 μM) in the presence of CGP54626 (2 μM). Right panel: Average frequency of sEPSCs before and after GABA (10 μM) in the presence of CGP54626 (2 μM) (*n* = 6, *P* > 0.05, compared to CGP54626 before GABA).**DOI:**
http://dx.doi.org/10.7554/eLife.09275.017

C) GABA_B_ receptor antagonist CGP54626 increases the frequency of sEPSCs in

VTA-DA neurons of mice and its influence on the effect of morphine.

We observed the effect of GABA_B_ receptor antagonist CGP54626 on the frequency of sEPSCs. The result showed that after the application of GABA_B_ receptor antagonist CGP54626, the frequency of sEPSCs significantly increased (Figure 14). The average frequency of sEPSCs was 2.7 ± 0.5 Hz before to 3.2 ± 0.4 Hz during 10-15 min after CGP54626 (2 µM) (n = 6, paired *t* test, *P* < 0.05, compared to control before CGP54626, right panel of Figure 3). In the presence of GABA_B_ receptor antagonist CGP54626, the effect of morphine on the frequency of sEPSCs disappeared (Figure 14). The average frequency of sEPSCs was 3.6 ± 0.6 Hz before to 3.6 ± 0.6 Hz during 10-15 min after morphine in the presence of CGP54626 (2 µM) (n = 6, paired *t* test, *P* > 0.05, compared to CGP54626 before morphine, right panel of Figure 14).

Author response image 2.Effect of GABA_B_ receptor antagonist CGP54626 on the frequency of sEPSCs and its influence of on the effect of morphine on the frequency of sEPSCs in VTA-DA neurons of mice.(**A**) Effect of GABA_B_ receptor antagonist CGP54626 on the frequency of sEPSCs in VTA-DA neurons. Left panel: Typical current traces of sEPSC before and after CGP54626 (2 μM). Middle panel: Time course of the frequency of sEPSCs before and after CGP54626 (2 μM). Right panel: Average frequency of sEPSCs before and after CGP54626 (2 μM) (*n* = 6, *P* < 0.05, compared to control before CGP54626). (**B**) Influence of GABA_B_ receptor antagonist CGP54626 on the effect of morphine on the frequency of sEPSCs in VTA-DA neurons. Left panel: Typical current traces of sEPSC before and after morphine (10 μM) in the presence of CGP54626 (2 μM). Middle panel: Time course of the frequency of sEPSCs before and after morphine (10 μM) in the presence of CGP54626 (2 μM). Right panel: Average frequency of sEPSCs before and after morphine (10 μM) in the presence of CGP54626 (2 μM) (*n* = 6, *P* > 0.05, compared to CGP54626 before morphine).**DOI:**
http://dx.doi.org/10.7554/eLife.09275.018

D) Presynaptic GABA_B_ receptor antagonist CGP36216 abolishes the effect of morphine on firing in VTA-DA neurons of mice.

As shown in raw firing traces (Figure 15) and time course of VTA-DA neuron firing (Figure 15), the presence of CGP36216 (100 µM) led to the disappearance of the effect of morphine (10 µM) on VTA-DA neuron firing. The average firing frequency of VTA-DA neurons was 2.2 ± 0.3 Hz before and 2.3 ± 0.2 Hz during 10-15 min after morphine in the presence of CGP36216 (n = 6, paired *t* test, *P* > 0.05, compared to CGP36216 before morphine, Figure 15).

Author response image 3.Influence of selective presynaptic GABA_B_ receptor antagonist CGP36216 on the effect of morphine on the firing in VTA-DA neurons of mice.(**A**) Representative spontaneous firing traces recorded before and after morphine (10 μM) in the presence of CGP36216 (100 μM). (**B**) Time course of spontaneous firing before and after morphine in the presence of CGP36216 (100 μM) (*n* = 6). (**C**) Average frequency of spontaneous firing before and after morphine (10 μM) in the presence of CGP36216 (100 μM) (*n* = 6, *P* > 0.05, compared to CGP36216 before morphine).**DOI:**
http://dx.doi.org/10.7554/eLife.09275.019

2) The aims of the study are well defined and the background is solid however I would add the following references. Concerning the disinhibition model, in addition to the Johnson and North paper I would add these two papers:

Kalivas PW (1993) Neurotransmitter regulation of dopamine neurons in the ventral tegmental area. Brain Res Rev 18:75-113.

White FJ (1996) Synaptic regulation of mesocorticolimbic dopamine neurons. Annu Rev Neurosci 19:405-436.

In the same line, I suggest including the study of “Bonci, A. and Malenka, R.C. (1999) Properties and plasticity of excitatory synapses on dopaminergic and GABAergic cells in the ventral tegmental area. J. Neurosci. 19, 3723-3730, when speaking about the inhibition of the glutamate terminals onto VTA DA neurons.

In the present revised version, we cited these three papers.

*3) There is no description of*
Figure 1
*in the text. The identification of DA neurons by the presence of an I*_*h*_
*current is not ideal and can be misleading. Did the authors consider using genetic methods for the identification?*

In this manuscript, all DA neurons we recorded were identified by both I_h_ currents and TH staining, except that mechanically dissociated DA neurons were identified only using I_h_ currents. We added an explanation for it as follows:

“Here, VTA-DA neurons were identified only using I_h_ currents (right of panel 1 of Figure 4) without TH staining because in dissociated DA neurons, it was difficult to fix the cell after recording for TH staining.”

Since the genetic methods for the identification of DA neurons is not available to us now, we do not use this method in the present manuscript.

*4) In*
Figure 2
*the authors should explain the choice of the morphine dose. Does it rely on a previous study, is it a physiological dose?*

The choice of morphine dose relied on previous studies. As morphine is an exogenous drug, it is a pharmacological dose, rather than a physiological dose. We added an explanation of it in the present revised version as follows:

“We observed the effect of morphine (10 µM) on spontaneous firing […] basically abolished by opioid receptor antagonist naloxone (38).”

5) Where is the figure showing that morphine has no significant effect on the amplitude of sEPSCs? Likewise the figure showing that the morphine has no effect on glutamate release from synaptosomes of the VTA is missing. Also missing is the figure showing that the CGP36216 could increase the firing rate of VTA-DA neurons.

In the present revised version, we added these figures.

*6) In*
Figure 2*, morphine had a strong excitatory effect on VTA DA neurons, which is no longer present on the dissociated DA neurons used in*
Figure 4*. The authors must refer to the results of*
Figure 2
*in the*
Figure 4
*description and detail the hypothesis underlying such difference.*

In the present revised version, we added a description to the text related to Figure 4 as follows:

“The mechanism underlying the promoting effect of morphine […] had a direct effect at glutamatergic terminals to promote glutamate release.”

*7) For*
Figure 6*, the authors should show all western blots (including the control) and not just a small part of it. Likewise, I would describe what chemical was used to label the GABA(B)R.*

In the present revised version, we showed all the western blots for Figure 6.

Rabbit anti-GABA_B_ R1 (Alomone labs) and IRDye 800CW goat anti-rabbit secondary antibodies (LI-COR Biotechnology) were used to label the GABA(B)R.

8) Morphine has been shown to induce CPP. In order to strengthen the behavioral part, the authors should perform a CPP test in order to observe the influence of CGP36216 on the effect of morphine reward. All four groups of the 2x2 factorial design must be reported.

We added this experiment.